# ANPL: Towards Natural Programming with Interactive Decomposition

**Di Huang**[1,†], **Ziyuan Nan**[1,3,†], **Xing Hu**[1], **Pengwei Jin**[1,3], **Shaohui Peng**[2],
**Yuanbo Wen**[1], **Rui Zhang**[1], **Zidong Du**[1], **Qi Guo**[1], **Yewen Pu**[4] **& Yunji Chen**[1,3⊠]

[1]SKL of Processors, Institute of Computing Technology, CAS
[2]Intelligent Software Research Center, Institute of Software, CAS
[3]University of Chinese Academy of Sciences
[4]Autodesk Research

https://iprc-dip.github.io/ANPL

## Abstract

Though LLMs are capable of generating plausible programs, it's challenging to interact with the LLMs further to revise the program, especially if the user's specific requirements are different from the initial proposal. In this paper, we introduce ANPL, an interactive programming system that ensures users can always refine the generated code towards their specific programmatic intents via structured decompositions. Borrowing the paradigm of sketching from program synthesis, an ANPL program consists of a set of input-outputs that it must satisfy, a "*sketch*" — control/data flow expressed in precise code (e.g. Python), and "*hole*s" — sub-modules to be implemented by the LLM specified with natural language. The user revises an ANPL program by either modifying the *sketch*, changing the language used to describe the *hole*s, or providing additional input-outputs to a particular *hole*, turning it into a sub-ANPL program that can be solved recursively. This workflow allows the users to offload programming burdens to the LLM as much as possible while retaining the ability to pinpoint and resolve bugs locally, without exposing the rest of the program to the LLM. We deploy ANPL on the Abstraction and Reasoning Corpus (ARC), a set of unique tasks that are challenging for state-of-the-art AI systems, showing it outperforms baseline programming systems that (a) without the ability to decompose tasks interactively and (b) without the guarantee that the modules can be correctly composed together. Additional evaluations on APPS, HumanEval, and real-world programming tasks have validated that the ANPL framework is applicable to multiple programming domains. We release the ANPL solutions to the ARC tasks as a dataset, providing insights into how humans decompose novel tasks programmatically.

## 1 Introduction

The rapid development of Large Language Models (LLMs) has made significant progress in the realm of code generation tasks [7–10, 20, 23, 31, 33, 34, 42, 44, 47, 71]. Compared to traditional approaches in program synthesis [21, 22] that were constrained to specific, narrow domains (*e.g.*, text manipulations), LLM-driven code generation can generate programs in a domain-general language (*e.g.*, Python), making it broadly applicable. While LLM-driven code generators can save much of programmers' time in compiling typical programs – programs that are similar to those present in their

---

[†]Equal contributions.
[⊠]Corresponding author. Contact: {huangdi, nanziyuan21s, cyj}@ict.ac.cn.

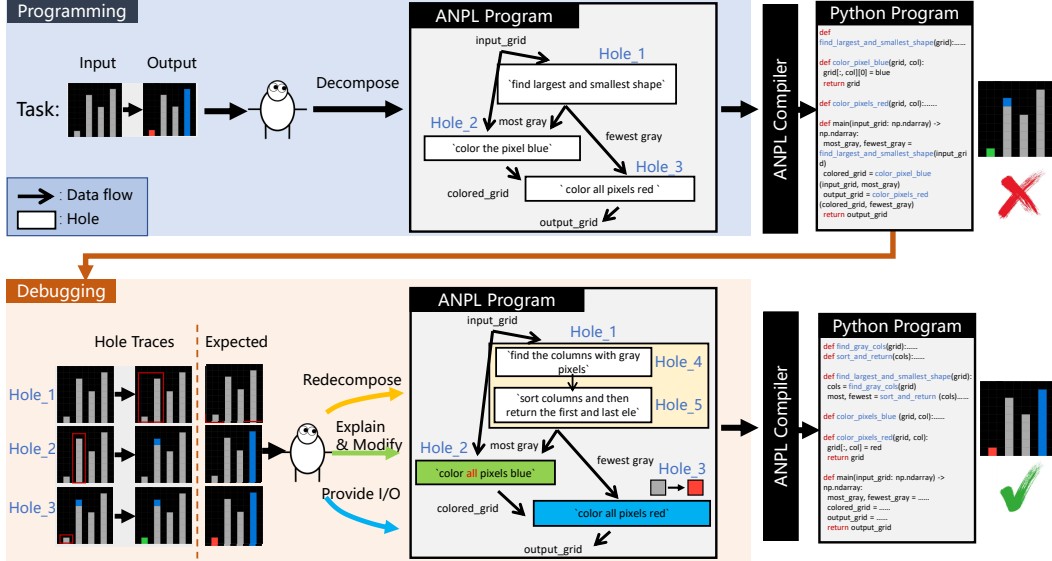

Figure 1: **ANPL overview**. Given a programming task (*e.g.*, input-output), the user decomposes it into an ANPL program, consisting of programmatically defined **sketch** (control/data flows, solid lines, programmed by the user and preserved by LLMs), and natural language **holes** (function modules, dashed lines, generated by LLMs). The ANPL compiler compiles it to a Python program and checks the Python code against the task's input-output. If incorrect, the compiler generates a debugging trace showing input-outputs for all the holes. The user can focus on debugging each hole independently, by either (1) decomposing it further, (2) explaining and modifying its natural language description, or (3) providing correct I/O by adjusting its traces.

training data, it is unlikely that an LLM can compile a user-specific program in 1-shot, without any additional user interventions [35, 62]. In software development, most code is not written perfectly on the first attempt, and users need to spend much effort on testing and debugging[46, 63]. Therefore, although using LLM for code generation can significantly reduce the cost of code writing in 1-shot, the subsequent interactions – editing and debugging an LLM-generated program – remain an open challenge. This challenge is severe in two aspects: (1) The LLM-generated program can be long and contain complex dataflows, making it difficult to make localized changes while ensuring the dataflows stay intact; (2) Language descriptions are inherently ambiguous to describe specific code edits.

We hypothesize that decomposition – breaking down a complex problem into simpler problems, a common practice in program synthesis [2, 48], interactive dialogue systems [30, 72], and cognitive science [16, 27] to name a few – during the programming and debugging interaction between a user and an LLM code generator can tackle these aforementioned difficulties. We propose the ANPL (Abstracted Natural Programming Language) system (Figure 1), which enables *stable* yet *effective* interactive editing/debugging to solve complex programming tasks that an LLM cannot correctly solve in 1-shot. The key design insight of ANPL is decoupling the dataflow "*sketch*" (something that requires little user effort) and the tedious module implementation "*hole*s" (automated with LLM). With ANPL, users have explicit control over the high-level task decomposition using the sketch, leaving the tedious work of function-level holes implementation for LLMs. Each hole is a stand-alone ANPL program, allowing the user to recursively decompose it further.

We evaluated ANPL by conducting a large-scale user study using the Abstraction and Reasoning Corpus (ARC) [14], a well-known corpus that consists of 400 unique tasks (in the training set) without predefined programmatic solutions. We recruited 19 Python programmers who interacted with our system for a total of 440 man-hours. Compared to prior works evaluating LLMs' code generation capabilities in interaction with real users Xu et al. [69](166 man-hours), Vaithilingam et al. [61](7.87 man-hours), ours is the most comprehensive evaluation up-to-date to the best of

our knowledge. We find that programmers interacting with ANPL perform significantly better than interaction with the vanilla LLM (75.0% tasks solved vs. 58.4%). The interaction between users and ANPL resulted in DARC, a dataset of programmatic Decomposition of ARC tasks, consisting of 300 instances of humans successfully decomposing ARC tasks to simpler tasks grounded in Python.

Specifically, this paper makes the following contributions:

- We present ANPL, a programming system that allows the user to explicitly manipulate dataflows while offloading the implementation of modules to an LLM, to decompose and ground programming tasks.
- We evaluate ANPL on a large-scale human study (440 man-hours) on a set of 400 ARC tasks, showing that programmers interacting with ANPL significantly outperform interaction with the vanilla LLM (GPT-3.5-turbo).
- We will release the collected programmatic decomposition dataset, DARC, to promote the development of research communities including LLMs, programming languages, human-computer interaction, and cognitive science.

## 2 Related Work

### 2.1 Code Generation with LLMs

LLMs have been applied to tackle the code generation problem, which is the task of automatically finding a program satisfying user intent expressed in natural language or other specifications [7, 20, 23]. For example, Codex [9], a GPT model finetuned on public code repositories from GitHub, demonstrates the potential of code generation with LLMs. With the idea of generating candidate programs and filtering, AlphaCode [34] samples a large number of candidate programs and filters using input-output examples. Chen et al. [8] and Key et al. [31] utilize formal specifications (*e.g.*, test cases and assertions) generated by LLMs to verify the generated programs. Synchromesh [47] retrieves few-shot examples from training sets and then checks syntax, scope, typing rules, and contextual logic when synthesizing programs. Some work focuses on generating code based on feedback. CodeGen [44] factorizes a single program description into multiple parts and generates programs in multi-turns. Chen et al. [10], Le et al. [33] and Madaan et al. [42] provide LLMs with feedback messages generated by an executor, a critic network, or LLMs themselves to improve performance. Although these efforts have achieved a high level of code generation quality, they do not incorporate user interaction within the process of code generation. Basically, these methods can be integrated into our compiler to gain better compiling results. Another type of work, like Parsel, also utilizes the concept of decomposition [71]. Different from Parsel, our work is an interactive system between users and LLMs, which allows users to debug programs through further clarifying and decomposing each module. Furthermore, we conduct a large-scale user study to evaluate the effectiveness of our system.

### 2.2 User Interaction with LLMs

Interactions between humans and LLMs are necessary because it is difficult for LLMs to accurately understand user intent and generate correct answers in one shot [36, 52, 67, 70]. Among them, InternGPT [39] finds it helpful to combine the advantages of non-language instructions (*i.e.* pointing instructions) and language instructions. Instead of pointing instructions, our system allows humans to specify control/data flow precisely with programmatic language while keeping details in natural language. Furthermore, we have conducted extensive human studies on the ARC benchmark, which sufficiently demonstrate the capabilities of our system.

### 2.3 Evaluation of LLM Code Generation Ability on Human Studies

Several works conduct a human study on code generation tools to survey generation accuracy and user experience [4, 38, 61, 69]. However, these works focus on analyzing *existing* tools such as Copilot. The purpose of our human study is to verify the effectiveness of our proposed interactive system, ANPL. Furthermore, ARC containing abstract problems is a harder benchmark than those in previous studies. Our human study takes 440 man-hours on this hard benchmark, greater than the usual 100 man-hours, producing reliable results.

Other related areas including programming with natural language and LLMs as an interpreter are discussed in the Appendix A.

# 3 ANPL: The Abstracted Natural Programming Language

ANPL is a programming system that allows users to program and debug with natural language. It is Python-like and compatible with the original Python language, which provides users with great programming flexibility. In principle, we can apply our system to any programming language and in this paper, we have opted for Python as our base language. In this section, we will introduce ANPL's design principle and how to specify a ANPL program and then debug it.

## 3.1 Design Principle

The main idea under ANPL is to decouple the description of *hole* and *sketch*, which frees users from the tedious coding of function-level *hole*s by allowing them to write the natural language and empowers them with the privilege of precisely setting *sketch* with few provided programming standards. Specifically, the *sketch* represents the control/data flow, and the *hole* represents the function module. This idea has been supported by the relevant experiences of early integrated systems like StreamBit [55] and SKETCH [54], which shows that users often have an idea about the general form of a solution (a high-level strategy) but feel it challenging to implement the program details. For programming with the help of LLMs, it offers the following advantages: 1) The user-specified *sketch* can convey user intent more precisely. 2) It enables users to trace bugs along the *sketch*, rather than aimlessly rewriting the entire natural language program and regenerating it. 3) The relationships between modules are explicitly represented by the *sketch*, thus making it easier for the compiler to ensure that other correct parts are not affected by debugging actions when users are debugging the erroneous sections of the program, which is not an easy task due to the unpredictability of LLMs.

## 3.2 Programming with ANPL

An ANPL program consists of a python-like *sketch*, and natural language *hole*s.

**Hole.** A *hole* implements a function module with a natural language description, which will be fleshed out by LLMs during the compiling process. Each *hole* specified with a natural language description quoted by quotation marks ` or """. When called, *hole*s should be organized by specifying its input-output variables, serving as the interconnections. To define a *hole*, users can either 1) define a *hole* as a sub-function with the function name, parameters, and descriptions, and then call the function with its function name and input-output variables, or 2) just define and call it with descriptions and input-output variables inline.

**Sketch.** A *sketch* is the control/data flow connecting different *hole*s, specified with a programmatic language. Users constitute the *sketch* by assigning names to variables and using them as *hole* parameters in a data flow graph. Besides, users can write complex control flows with programmatic keywords (*e.g.*, for, while, if) similar to that in Python to get more intricate ANPL programs.

An ANPL code example can be found in Figure 2, Figure 3, and Appendix D.1.

## 3.3 Debugging with ANPL

As an ANPL program is already decomposed into *hole*s connected by the *sketch*, debugging over ANPL is targetted, where users can accurately locate bugs and make modifications to the relevant ANPL program.

**Tracing.** Tracing helps users to locate bugs. With the clear and explicit definition of *sketch* in ANPL, users are allowed to detect bugs by checking the execution trace between high-level *hole*, which is not easy in previous work [71]. Users can either use the preset "Trace" option in ANPL or just directly insert breakpoints in the code with `print`.

**Providing input-output constraints.** Users can provide input-output constraints, a typical gold standard for correctness, and ask the ANPL compiler to resynthesize the program to ensure the

```
def compiling(ANPL):
  while "ANPL program has a hole"(ANPL):
    hole = "return the first hole in ANPL program"(ANPL):
    generated_code = LLM(ANPL, hole.descriptions, hole.IOs)
    dependency_graph = "analyze generated code, return dependency graph with topology
    ↪  sort"(generated_code)
    entry_nodes = "return all entry nodes of the dependency graph with topology sort"(dependency_graph)
    if "the hole has been named by the user"(hole):
      new_dependency_graph = "remove unrelated nodes in the dependency graph"(dependency_graph, hole)
      ANPL = "fill the hole with generated code according to the dependency graph"(ANPL, generated_code,
      ↪  new_dependency_graph)
    elif len(entry_nodes) == 1:
      ANPL = "fill the hole with generated code according to the dependency graph"(ANPL, generated_code,
      ↪  dependency_graph)
    else:
      # The dependency graph has multiple entry nodes, re-generate the hole
      continue
  return ANPL

def compilingDiff(ANPL_old, ANPL_new):
  if "ANPL program has a hole"(ANPL_new):
    ANPL_new = compiling(ANPL_new)
  ANPL = "merge two dependency graphs according to the merging principle of intention: user_new >
  ↪  user_old > LLM_old > LLM_new"(ANPL_old, ANPL_new)
  return ANPL
```

Figure 2: The pseudo-code of ANPL compiler, consisting of the direct compiling process and the differential compiling process.

correctness of the compiled program. Each resynthesis will arise 1 API call with a batch size of 10 of the LLM.

**Editing.** ANPL allows users to debug as they would in traditional programming languages. Specifically, users can edit their code in four ways: 1) They can recursively decompose the original *hole* into lower-level *hole*s connected by the *sketch*. 2) Reversely, they can abstract low-level *hole*s and the corresponding *sketch* to a higher-level *hole*. 3) They can explain and modify the natural language descriptions of *hole*s. 4) They can change the *sketch* by coding (*e.g.*, changing the input-output variables of *hole*s).

In addition, ANPL has various syntactic sugars, for instance, implementing a recursion with natural language by using the Y combinator, see Appendix D for details.

## 4 The ANPL Compiler

One of the crucial tasks that a compiler must undertake is to gain the trust of its users - it should accurately reflect the underlying code and provide users with the possibility of debugging. To achieve this task, the ANPL compiler mainly focuses on two problems: 1) in the program generation process, how to generate a reliable executable program while still preserving the control/data flows specified by users, and 2) in the debugging process, how to allow users to modify and regenerate the erroneous parts of the program while preserving functionalities in the rest of the programs.

### 4.1 Compiling ANPL

**Preserving the *sketch*.** To ensure the consistency of control/data flows before and after compilation, we first extract control/data flow information from the *sketch* and individual natural language descriptions from the *hole*s separately. Then, we traverse and implement these *hole*s while ensuring the integrity of the *sketch*. Note that while LLMs take in the entire ANPL program as context to implement the program of its *hole*s, the *sketch* is fixed during program generation.

**Generating the *hole*s.** LLMs iterate over and fill the *hole*s one by one in their appearing order. When filling, LLMs can automatically create a name for the *hole* and decompose it into sub-functions. The main problem is that LLMs cannot implement *hole*s reliably and may generate useless functions due to their unpredictability. The proposed solution begins with analyzing and clarifying the dependency relationships among the generated functions, and then identifying which function serves as the implementation of the target *hole*. The fundamental principle is that if the user explicitly names the *hole*, then any generated function with the same name as the user-specified one should be considered as the implementation of the *hole*, while other functions that it depends on are regarded as sub-functions of the *hole*; And otherwise, the function that does not depend on any other functions should be considered as the implementation of the *hole*. In the implementation, we first construct a dependency graph from the generated functions and find its entry nodes (nodes without any dependency on it) of the graph with topology sort. Then, we identify the target function based on three situations: 1) The user has named the *hole* and we just match it with the entry node's name and remove unrelated nodes. 2) There is only one entry node named by LLMs and we fill the target *hole* with it. 3) There are multiple entry nodes which means that there are multiple independent groups of functions and we cannot judge which one should be filled into the target *hole*, and thus we instruct LLMs to re-generate the implementation.

Finally, we get a compiled Python program that preserves user-specified *sketch* and has a well-organized implementation of *hole*s.

## 4.2 Compiling Differences

During the debugging process, users generate an updated ANPL program. The problem is how to modify the erroneous parts while preserving the rest of the compiled Python program.

We observe that this process can be modeled as the merging of dependency graphs. Specifically, the old ANPL program corresponds to one dependency graph, and the new program debugged by users serves as another new dependency graph, and the merging process. For merging, we propose three assumptions: 1) the user's current intention takes priority over previous intentions, 2) the LLM's previous intentions take priority over the current intention, and 3) the user's intentions take priority over the LLM's intentions. Assumption 1) and Assumption 3) are obviously valid: It is precisely because the user wants to modify their previous intention or correct the compiled result with LLM errors that they will perform debugging operations. Assumption 2) holds because we want to minimize the impact of debugging on the generated program: When the user's intention does not involve the code, the code should not undergo any changes due to LLM's compilation.

Based on these assumptions, we can derive a fundamental principle for the priority of dependency graph merging: the user's current intention > the user's previous intention > the LLM's previous intention > the LLM's current intention.

# 5 Human Studies

We conducted a user study on 400 ARC training tasks to evaluate the effectiveness of ANPL compared to the original ChatGPT (GPT-3.5-turbo). We hypothesized that if ANPL can help people to program with LLMs, people should be able to solve more tasks and get the right program with less effort (*i.e.* time consumption) on solving and programming tasks.

## 5.1 Settings.

**Stimuli.** We use the ARC as our programming tasks. The ARC benchmark [14] was proposed to provide a standardized set of tasks that can be used to evaluate key aspects of human general intelligence, such as abstraction, generalization, object categorization, and so on [5, 13, 25, 29, 32, 40, 59, 60]. The primary objective of ARC is to require individuals to deduce a consistent procedure based on a limited number of abstract input-output examples, and then apply this procedure to generate a previously unseen output for a new input, as depicted in Figure 1. It is found that ARC is a representative programming dataset because 1) humans tend to solve them by coming up with

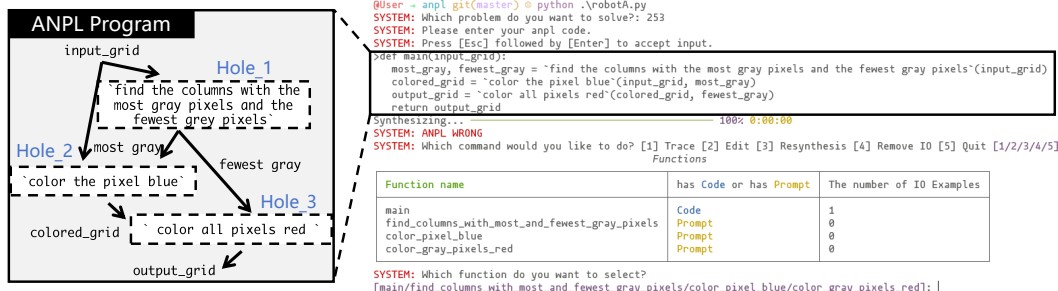

Figure 3: **ANPL user interface**. Participants utilize this programming interface to write and debug ANPL programs. The figure shows the ANPL program code entities along with corresponding *sketch* and *holes* diagrams. For a more detailed user interface, please refer to the Appendix E.3.

programs [1, 29][1], and 2) the solution of ARC cannot be found on the internet, which make its problems unique enough for LLMs (and participants).

**Participants.**    We recruited 19 primary Python programmers from our institution. Due to the difficulty of solving ARC problems by interactive programming, we first recruited more than 19 participants and then filtered out 19 of them by asking each person to complete ∼10 problems, checking if they can correctly operate the ANPL system, and asking if they wanted to continue. We paid an average of ∼ $8.80 USD per hour – a standard programmer's wage within our institution.

**Design.**    1) Each participant is randomly assigned a set of non-overlapping ARC problems. The number of problems is different per participant depending on their preferred commitments. 2) There are two systems: System **A**, the ANPL, and System **B**, the original ChatGPT (GPT-3.5-turbo). Participants can interact with System B arbitrarily, including inspecting and changing Python programs manually. However, they cannot implement a whole Python function from scratch without interacting with System B. 3) As the programming level of participants can have a significant impact on their ability to solve ARC tasks, for each task, each participant used both System A and System B, presented in random order. 4) The participants were informed that the first interaction is crucial in order to minimize the impact of subsequent interactions. The programming interface of System A is shown in Figure 3 and System B shares a similar interface to System A.

**Procedure.**    Given a set of tasks, participants can decide on the order in which to complete them, as they may need to start with simpler ones first. Participants are asked to solve the problem by programming (or interacting) with System A (or B) to get a Python program to solve the task. A soft time limit is 30 minutes which means that participants are suggested to give up after a 30-minute try. The held-out test input-output examples provided in ARC are used to check the correctness of the generated Python program, and participants can use additional input-output examples to debug their program. Participants are shown some prompt templates for using System B, while no prompts are needed in System A.

## 5.2   Results

**Participants solve more tasks with ANPL.**    First, we examine the problem-solving rate of ARC tackled by the participant when using different systems, shown in Figure 4. Specifically, we compare System A (ANPL), System B (GPT + interaction, *i.e.* ChatGPT), System C (ANPL without interaction), and System D (vanilla GPT), where Systems C and D represent the solving rate of one-shot generation without further interaction using Systems A and B, respectively. Results show that System A mean = 75.0%, 95%CI = [70.8%, 79.1%], B mean = 58.4%, 95%CI = [53.4%, 63.3%], C mean = 23.5%, 95%CI = [19.4%, 27.5%]), and D mean = 16.8%, 95%CI = [13.2%, 20.6%], where CI denotes confidence interval. Furthermore, we conduct paired t-tests on these systems, all of which are shown significant ($df$ =

---

[1]the top results on a Kaggle competition of ARC are all based on program synthesis techniques.

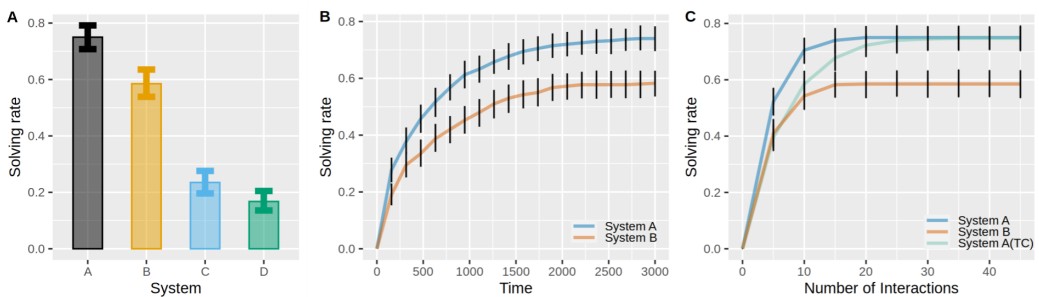

Figure 4: Results of problem-solving rates. A: Solving rate of four systems. B: The relationship between solving rate and time consumption. C: The relationship between solving rate and number of interactions. Trace Calculated(TC) means the trace mode is considered into interactions.

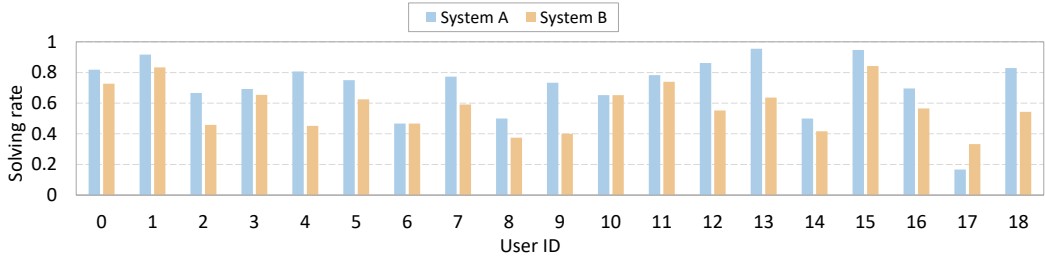

Figure 5: User differences. The solving rate of different users using System A and System B.

399, $p < .0001$): A-B: t=7.606, 95%CI=[12.2%,20.8%]; A-C: t=20.583, 95%CI=[46.6%,56.4%]; A-D: t=23.594, 95%CI=[53.4%,63.1%]; B-D: t=13.646, 95%CI=[30.0%,40.0%]; B-D: t=16.911, 95%CI=[36.9%,46.6%]; C-D: t=3.152, 95%CI=[2.5%,11.0%]. **In conclusion**: 1) With large confidence, ANPL performs the best and allows users to solve more problems, with a solving rate achieving 75.0% on average, while B achieves 58.4% (↑ 28.25%). 2) Programming with interaction (Systems A and B) is always better than programming without interaction (Systems C and D). 3) Even for one-shot code generation, ANPL without interaction (System C) performs better than vanilla GPT (System D).

**ANPL saves user effort.** Next, we measure the user effort on different systems. Figure 4 illustrates the relationship between solving rate and time consumption, as well as the relationship between solving rate and the number of interactions. We find that compared with GPT, ANPL exhibits a rapid growth curve, implying a significant advantage in terms of efficiency in problem-solving. Furthermore, to estimate the coding effort, we conduct a comparative analysis of the average lines of code between ANPL and the generated Python code. Some ARC task is simple and one of the main reasons for the time cost on simple tasks is that during programming, much time is spent waiting for GPT's response. The average waiting time is 14.6% for System A and 23.5% for System B. Besides, the users were instructed to solve the tasks as fast as possible and they chose not to program directly when using System A because it would require more effort. Another way to analyze user effort is to calculate the lines of code. We find that the [min, max, mean] of lines of ANPL is [2, 76, 10.29] while the one of Python is [4, 113, 26.47], and note that writing natural language is simpler than coding. We also demo the ANPL's application on projects including MAGIC card game and LS-8 CPU with 67% lines of code reduction (See Appendix F.4).

**User differences.** The characteristics of the ARC benchmark lead to significant variations in user performance. In order to measure such differences, we conducted an anonymous statistical analysis of users' solving rates while utilizing Systems A and B. Figure 5 shows that different users achieve variant solving rates. The (mean, max, min) solving rate achieved by users when using System A is (71.1%, 95.5%, 16.7%) with variance 0.038, while System B is (57.2%, 84.2%, 33.3%) with variance

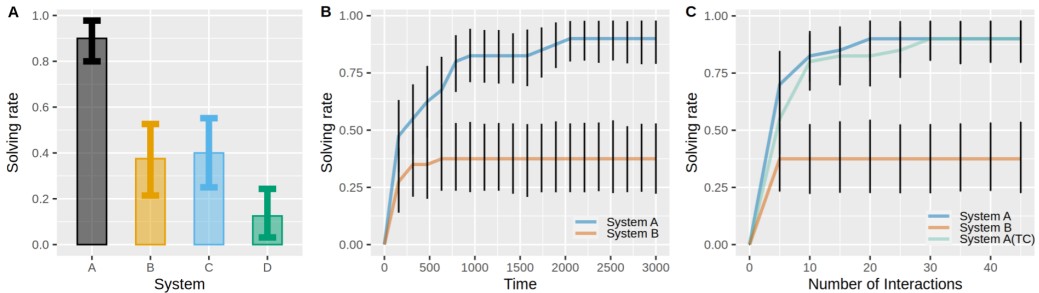

Figure 6: Results of problem-solving rates when users can not see generated Python code and only interact with user interfaces and IO feedback. A: Solving rates of four systems. B: The relationship between solving rate and time consumption. C: The relationship between solving rate and number of interactions. Trace Calculated(TC) means the trace mode is considered into interactions.

0.022. This illustrates that although System A surpasses B in most cases and on three metrics, it exhibits greater instability, suggesting the presence of a certain learning cost compared with B.

**ANPL performs even better for programming beginners.** In order to simulate the situation where users are unable to understand Python code (programming beginners), we designed a new set of experiments with System A and System B. Neither system provides users with specific Python code; instead, users are required to complete tasks only based on user interfaces and IO feedback.

We selected four users for the experiment. To mitigate the potential impact caused by variations among users. Each user was given a set of 10 problems, which they had previously answered correctly using both System A and System B. In total, there were 40 problems administered. Results (Figure 6) show that System A mean = 89.9%, 95%CI = [79.4%, 97.8%] B mean = 37.2%, 95%CI = [21.6%, 52.5%], C mean = 40.2%, 95%CI = [24.3%, 55.6%], D mean = 12.5%, 95%CI = [2.9%, 25.0%]. With large confidence, ANPL performs the best and allows users to solve more problems, with a solving rate achieving 89.9% on average, while B only achieves 37.2% (↑ 141.67%).

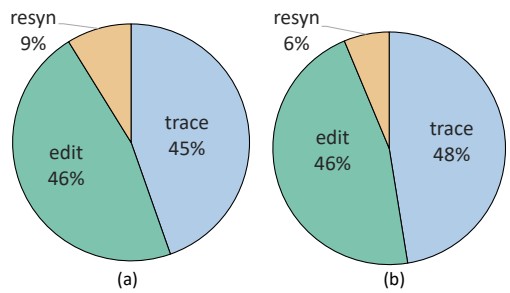

Figure 7: The proportion of three interaction modes. (a) Time. (b) Frequency.

**Percentage of different interaction modes.**
It can be observed from Figure 7 that among all the available options, the majority of users prefer the "edit" action, accounting for 46.2% total number of interactions and 46.6% total time cost, followed by the "trace" action, which constitutes 47.5% total number of interactions and 44.6% total time cost. This trend aligns with the typical programming workflow, where a smaller portion of time is dedicated to coding, while a significant portion is allocated to testing and debugging.

**The influence of the system order.** We denote the order as "A-B" ("B-A") meaning that the user first use System A (B) and then System B (A). With "A-B", the solving rate of System A is 75.5% (151/200) and System B is 62.5% (125/200). With "B-A", the solving rate of System A is 74.5% (149/200) and System B is 54.5% (109/200). We conclude that System B's performance increases 8% under the influence of System A, while System A's performance is more stable. This may be because System A tends to encourage users to approach problem-solving in a *sketch* and *hole* decomposition manner, and this manner is subconsciously applied to the subsequent System B. Therefore, we believe that "thinking in decomposition" itself can bring about certain improvements, and A leads users to this thinking manner. Additional results are shown in Appendix F.2.

### 5.3 DARC: A Recursive Decomposition Dataset of ARC Tasks

DARC contains all recorded interactions between participants and both the ANPL system and GPT-3.5-turbo on all 400 ARC tasks. 300 of the 400 tasks can be successfully decomposed and grounded in Python using ANPL (Figure 8), with an average of 2.7 *hole*s. Due to the characteristics of ARC, the data contained in DARC is not the only solution, and different users have a significant impact on the dataset. Still, the value of our dataset is tremendous, as it contains: programmatic decompositions of ARC tasks by humans using ANPL, Python code for 300 ARC tasks, and fine-grained interaction histories between users and a code-generating LLM.

## 6 Conclusion

In this paper, we propose ANPL, a programming system that enables users to program and debug with natural language with interactive decomposition. The system was evaluated on a large-scale user study on the 400 ARC tasks. Experimental results show that with large confidence, programmers interacting with ANPL perform significantly better than interaction with the vanilla LLM (75.0% tasks solved vs. 58.4%). Furthermore, we will release the collected programmatic decomposition DARC dataset, which provides valuable insights into how humans decompose programmatic tasks into simpler ones using natural language and input-outputs.

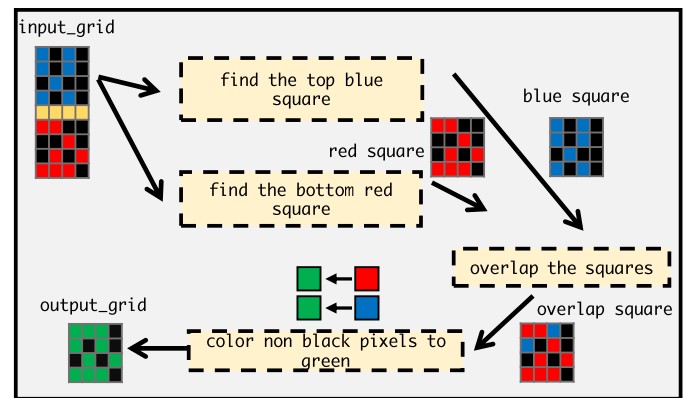

Figure 8: An example in DARC.

## 7 Limitations

Our limitations and future work mainly focus on two aspects: the naturalization of LLM responses and library learning. Specifically, 1) the ANPL system should return easily understandable natural languages instead of Python code as a response to humans, and 2) how to build a natural language library community through library learning [19, 66] to allow users to share and reuse programs made by each other. Please see our detailed limitations and broader impact in the Appendix B & C.

## Acknowledgments and Disclosure of Funding

We thank Tianyun Ma and Yutong Wu for their effort in submission and rebuttal. Tianyun helped us analyze data and Yutong conducted the automatic code generation experiment.

This work is partially supported by the NSF of China (under Grants 62002338, 61925208, 62222214, 62102399, U22A2028, U19B2019), the National Key R&D Program of China (under Grant 2021ZD0110102), CAS Project for Young Scientists in Basic Research (YSBR-029), Youth Innovation Promotion Association CAS and Xplore Prize.

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

# Appendix

## Table of Contents

# A  Additional Related Work

## A.1  Programming with Natural Language

Due to the low learning requirements, programming with natural language has been seen an attractive programming mode since 1960s [6, 15, 17, 24, 37, 43, 49]. The relevant domains encompass semantic parsing [28, 65], language grounding [57, 58], and so on. Among these works, Wang et al. [64] is the most similar one which let users build complex voxel structures by defining alternative, more natural syntax, and increasingly complex natural concepts, starting from a core programming language. However, they are restricted by the pre-defined natural language specifications and domain-specific languages while this paper focuses on unconstrained natural language and general-purpose languages like Python. Furthermore, it should be noted that the interactive processes involved in these studies rely on manually annotated data, whereas our system operates in a truly debugging fashion for programs composed of natural language.

## A.2  LLMs as an Interpreter

LLMs have been embedded in programs as an interpreter, due to their capabilities of common sense question answering and simple natural language reasoning [50, 51, 68]. For example, Cheng et al. [12] leverages LLMs to generate programs for questions like "Is Mexico North America?" with Codex API calls like $f(NorthAmerica?, Mexico)$ and then answer it by executing and prompting. Dohan et al. [18] composes LLMs into a probabilistic programming framework, which allows control flow and dynamic structure. Different from these works, we focus on leveraging LLMs to implement natural language modules into executable programs. We take the utilization of taking LLMs as parts of our generated programs as future work.

# B  Detailed Limitations and Future Work

Though our system has shown excellent performance revealed by the large-scale human study, there are still three main limitations to our current system. The first limitation is the response of LLMs. ANPL gives users a detailed implementation of each hole in Python and asks them to further debug. This requires users to read Python code when editing ANPL programs. LLMs may employ certain APIs that users may not be familiar with, resulting in the implementation of a function that deviates from the users' intended approach. Consequently, comprehending Python code becomes more challenging. Additionally, when LLMs automatically break down a function into sub-functions, identifying the specific sub-function containing a bug becomes difficult for users. To enhance the user-friendliness of ANPL and alleviate the burden on users in terms of their code capabilities, **how can ANPL provides concise summaries of each function in easily understandable natural language** should be studied. These summaries would enable users to identify any misinterpretations or incorrect implementations based on the corresponding descriptions, allowing them to modify their natural language descriptions accordingly without the need for complete redrafting.

Another limitation is the absence of comprehensive natural language libraries. In the context of code generation using LLMs, the quality of the prompt and the accompanying description assumes paramount importance. However, the creation of effective natural language descriptions necessitates considerable expertise in prompt engineering. In order to mitigate this issue, **a natural language library should be established**. The natural language content within this library is derived from two primary sources: library learning methods [66] and user contributions, and users can share and reuse natural language and corresponding implementations made by each other.

So far, ANPL has been limited to generating Python code for solving ARC problems through communication with users. However, as mentioned earlier, some ARC tasks cannot be fully addressed with Python programs alone and require the use of neural modules like object detection. Therefore, we need to **integrate ANPL with neural modules**, similar to works including Cheng et al. [12], Shen et al. [52], Surís et al. [56]. This integration would further enhance the capabilities of the ANPL compiler and expand the range of tasks that ANPL can handle. Additionally, the application scope should not be confined solely to the ARC dataset but should extend to multiple domains, such as chip design, program writing, and robot control. For example, Cheng et al. [11] automatically designed a CPU with Binary Speculation Diagram (BSD), yielding modular micro-operations that can be utilized by ANPL compiler and makes it possible to use ANPL-designed CPUs in the future. However,

due to the unique characteristics of ARC itself and the limited availability of human resources, it is well-suited for conducting user-programming experiments and human study reports. Thus, we have chosen ARC as the main platform for conducting our experiments and presenting our findings.

## C   Broader Impacts

On one hand, ANPL sheds light on the human-computer interaction paradigm by making the interaction more stable and reliable through low-cost predefined programming conventions. On the other hand, ANPL has the potential to promote the development of the programming field and broaden the scope of programming applications. For example, ANPL enables users to program and debug with natural language which can significantly lower the programming barrier. Furthermore, the proposed DARC dataset reveals how humans systematically decompose complex problems into simpler ones when faced with logical problems similar to ARC. This could provide further insights to cognitive science and foster advancements in related fields.

However, ANPL also raises safety challenges by producing code that is unaligned with user intent and can be misused. We refer readers to the broader impacts and hazard analysis discussed comprehensively by Chen et al. [9] as the basic component of the ANPL compiler is the LLM. Note that, compared to Chen et al. [9], a more significant concern with the usage of ANPL is its lower barrier to entry, allowing individuals with limited programming experience to generate code. This may result in code generated by ANPL lacking the scrutiny and maintenance of experienced personnel, making it more susceptible to misuse and thereby leading to more severe issues related to code security. Besides, since these users may have limited exposure to open-source communities such as GitHub, models trained on corresponding data may face greater challenges in user alignment.

# D ANPL Details

## D.1 An ANPL Code Example

Figure 10 is an ANPL code example for the task shown in Figure 9. In this example, users first decompose the task into `"Change the input into four new arrays based on the central dividing line in`
↪ `the x and y directions"` and `"Find an array that doesn't have just one color"`. Then, users can either define a *hole* by its name like `seperate_input` with corresponding parameters, or they can just code a piece of natural language description like `"Find an array that doesn't have just one color"` and set the *sketch* by specifying its input-output variables.

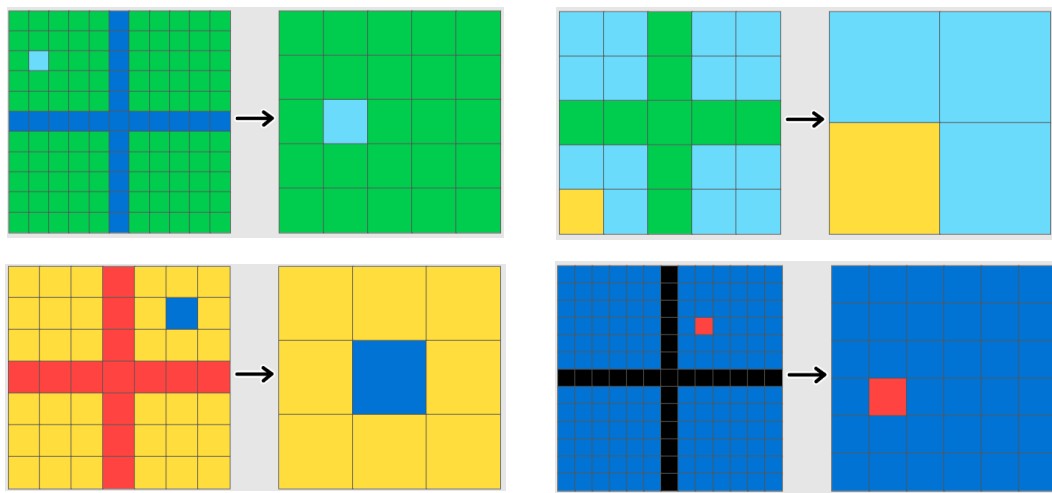

Figure 9: Task 64 of ARC.

```python
def seperate_input(input):
    "Change the input into four new arrays based on the central dividing line in the x and y directions"

def main(input):
    inputs = seperate_input(input)
    output = "Find an array that doesn't have just one color"(inputs)
    return output
```

Figure 10: An ANPL code example for task 64.

## D.2 Syntactic Sugar

**Recursion.** ANPL employs a hole within the function's own body to implement recursion, following the thought of Y Combinator. A function can indirectly invoke itself by passing its own reference to a hole within its own body.

Figure 11 is an example demonstrating the use of hole-driven recursion with the Flood Fill algorithm. In this example, the floodfill function is passed as an argument to the hole `"apply floodfill to`
↪ `adjacent pixels: above, below, right, and left"`. The semantics of the hole suggests that floodfill will be applied to the adjacent pixels, establishing an indirect recursion.

## D.3 Implementation Details

ANPL interacts with the LLM via the prompt shown in Figure 12. In the prompt, the **code** section will be substituted with all the executed codes up to that point, and the **hole** section will be replaced

```python
def floodfill (grid, i, j):
    if "is outside the valid grid area"(grid, i, j) and grid[i, j] != black:
        return grid
    else:
        return "apply floodfill to adjacent pixels: above, below, right, and left"(grid, i, j, floodfill)
```

Figure 11: Recursion in ANPL.

with the designated function name of the hole along with the natural language description given by the user. During the initial user input and function editing, it goes through a sequence of five attempts, starting with a temperature parameter of 0 and incrementing it by 0.1 with each try until it succeeds. In the resynthesis stage, ANPL requests the underlying LLM to produce 10 potential completions for each prompt. The text that ChatGPT generates will be subject to a maximum token constraint of 1024.

```
# system prompt
As a pythonGPT, your task is to complete the unimplemented functions in the given python code,
which are referred to as "holes" and are labeled as _hole0, _hole1, _hole2, and so on.
Your implementation should align with the code and documentation using Python.

# user prompt
```python
{code}
```

The function needs to be given a new name. Markdown format should be used to return it.
```python
{hole}
```
```

Figure 12: Prompts used in ANPL.

# E   Human Studies

## E.1   Questionnaire

We conducted a survey to investigate users' programming abilities, LLM usage experiences, evaluations of system A and system B, as well as the perceived importance of various functionalities within system A. The detailed questions are as follows:

1. How would you rate your programming skills?
   1: Non-programmer
   2: Beginner, struggles with solving LeetCode medium-level problems
   3: Familiar with a programming language, understands basic data structures and algorithms, able to solve some LeetCode medium-level problems
   4: Proficient in common data structures and algorithms, capable of solving many LeetCode medium-level problems
   5: Skilled in data structures and algorithms, capable of solving LeetCode hard-level problems

2. How familiar are you with the Python language?
   1: No exposure to Python.
   2: Have used Python, familiar with basic syntax, but rarely used in daily activities.
   3: Occasionally use Python to write simple scripts, not familiar with Python libraries such as NumPy and PyTorch.
   4: Proficient in Python features, frequently use Python, and familiar with some libraries.
   5: Mastery in Python and proficiency in using common libraries.

3. Have you used language models to generate code before?

4. How do you perceive the difficulty of ARC questions? (1: Very easy - 5: Very hard)

5. Do you find System A (ANPL) useful? (1: Very dissatisfied - 5: Very satisfied)

6. Do you find System B (natural language) useful? (1: Very dissatisfied - 5: Very satisfied)

7. How important do you consider the Trace feature of System A? (1: Very unimportant - 5: Very important)

8. How important do you consider the Edit feature of System A? (1: Very unimportant - 5: Very important)

9. How important do you consider the Resynthesis feature of System A? (1: Very unimportant - 5: Very important)

The results are shown in Figure 13. Participants in our human study are primary Python programmers and half of them are not familiar with code generation with LLMs. The average score of system A is 4.05, significantly greater than system B which scores 2.58. System A achieves not only a higher solving rate but also a better user experience than System B. Besides, most users find tracing and editing useful while resynthesizing less important, which shows the limitation of LLM code generation and indicates the importance of introducing user interaction in solving complex tasks like ARC.

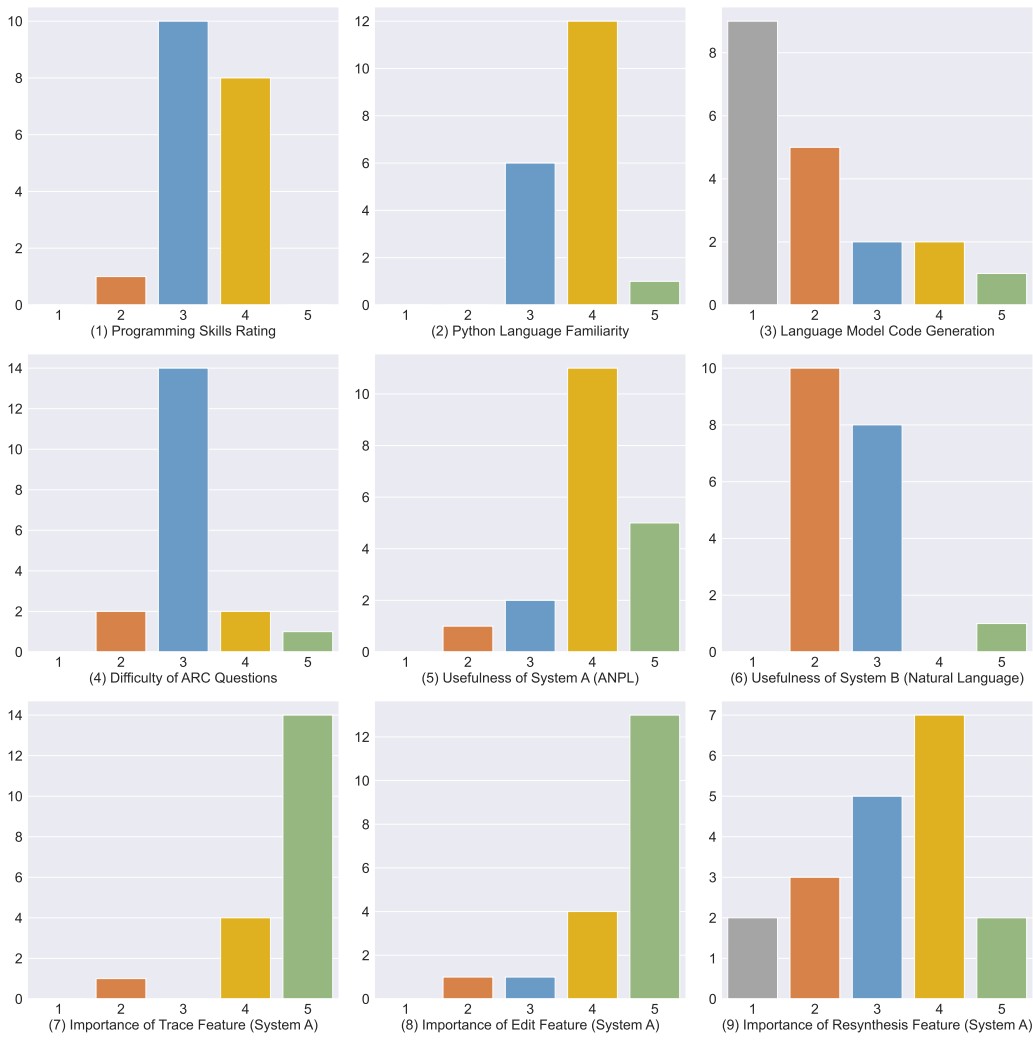

Figure 13: Questionnaire Analysis. The average score of System A is 4.05, and System B is 2.58.

## E.2 Tutorials

The task requires you to solve ARC (Abstraction and Reasoning Corpus) tasks, each composed of several input-output pairs. These pairs maintain a uniform pattern and are structured as color grids with ten distinct colors: black, blue, red, green, yellow, grey, pink, orange, teal, and maroon.

The goal is to deduce patterns from the input-output pairs and communicate your solution to the system. In this experiment, you will interact with two systems: System A and System B. System A accepts ANPL inputs, whereas System B functions on full natural language. Both systems aim to generate Python code that transforms the input into the expected output.

Each task requires working with both systems in a specific order provided in the task assignment. To begin, find the ARC task solution independently (solve the task in your mind, i.e. not with Python). With a solution in hand, activate the system, which will initiate a timer.

Your target is to instruct the systems to generate accurate Python code based on your solution. We will evaluate the program strictly on the test input and output, yet it's essential for you to confirm that your program is capable of successfully handling all the training input and output. Once the correct code is produced, the system will automatically deactivate. Perseverance is crucial, but if the system fails to generate the accurate code within 30 minutes, you're permitted to terminate the process. If a task proves overly challenging at any point, it's acceptable to stop prematurely.

System A is comprised of three primary operations. The *trace* operation allows for a function name to be entered, which triggers the program to run on a test input and displays all the input and output data of the selected function. The *edit* operation allows for direct changes to a function's body, including alterations to the sketch and hole. This operation has four sub-operations:

- Splitting the original function into multiple holes linked by the sketch.
- Turning the original code into a hole and attempting code generation.
- Changing the natural language description associated with the hole.
- Modifying the sketch while maintaining the generated hole.

The *resynthesis* operation requires the user to provide correct input and output examples. The system then generates numerous functions and tests in which one meets these examples. The provided examples are kept for future use, and multiple sets of examples can be provided by the user.

System B incorporates two operations. The *chat* operation allows the user to interact with the system using natural language, leading to code generation or modification based on these descriptions. The *remove history* operation allows the user to select and delete some historical conversation data[2].

## E.3 User Interface

The user interface consists of 4 components: tracing operation, editing operation, resynthesizing operation, and a grid editor.

**Tracing.** The tracing operation has three panels, namely function selection, visual IO, and textual IO, see Figure 14. The function selection section allows users to choose from a list of available functions eligible for tracing. After selecting a function, IOs are shown to users within the visual IO and the textual IO panels, where the visual IO panel visualizes the IO into grids and the textual IO panel prints the IO as NumPy arrays.

**Editing.** The editing operation has three panels: function selection, function editing, and code synthesis, see Figure 15. The function selection panel serves the same purpose as the one in the tracing operation. After selection, users can modify ANPL code in the function editing panel and submit it to the LLM for code generation. They can then view the code generation progress in the code synthesis panel.

---

[2]Note that participants were free to delete or modify their chat content and they can even write an ANPL program and then "compile" it by hand when using System B. Also, we did not explicitly forbid the users to write Python code in our experiment so they can write Python code if they want.

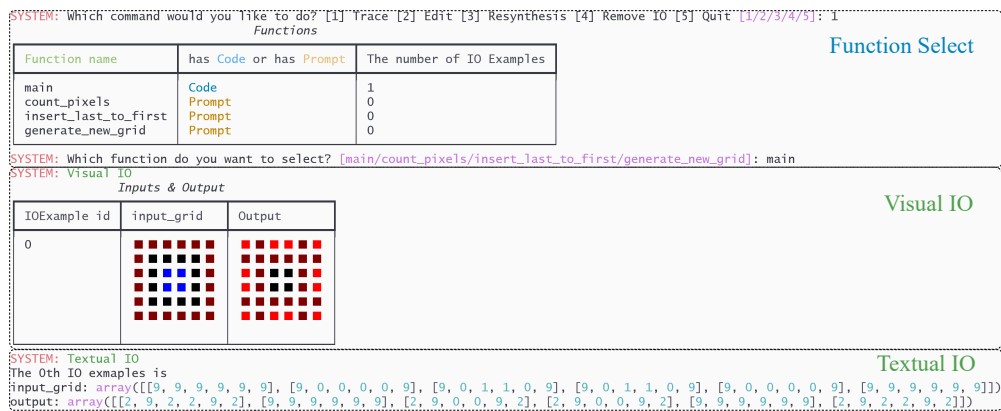

Figure 14: The tracing operation. Users can check the execution trace between high-level holes.

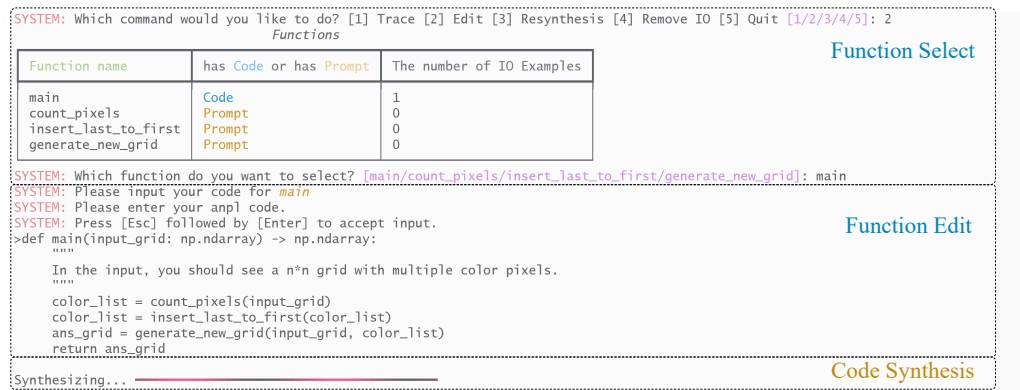

Figure 15: The editing operation. Users can edit the existing ANPL program through further decomposition or just modifying the code or natural language.

**Resynthesizing.** The resynthesis operation has three panels: function selection, IO entering, and code synthesis, see Figure 16. The function selection panel and the code synthesis panel serve the same purpose as the ones mentioned above. The IO entering panel enables users to constrain the programs generated by the LLM by providing IOs. Specifically, LLM generates a set of 10 candidate Python programs, and subsequently selects the program(s) that satisfy the given IO constraints as the compiled program.

**Grid editor.** In order to facilitate the transition for users between visual IO (*i.e.* colored grids) and textual IO (*i.e.* NumPy array), we have implemented a grid editor (Figure 17). This editor consists of the following elements:

1. Resize: Allows users to specify the dimensions of the grid.
2. Generate: Generate the corresponding grids by inputting a Numpy array.
3. Reset: Reinitializes the current grid to its initial state.
4. Copy: Converts the current grid into a Numpy array and copies it to the clipboard.
5. Graphical editing area: Provides three operations, including edit, select, and flood fill, along with 10 different colors, for direct editing of the colored grids.

Note that this editor was provided as a supplementary tool *to all systems*, so it has no impact on the final experimental results.

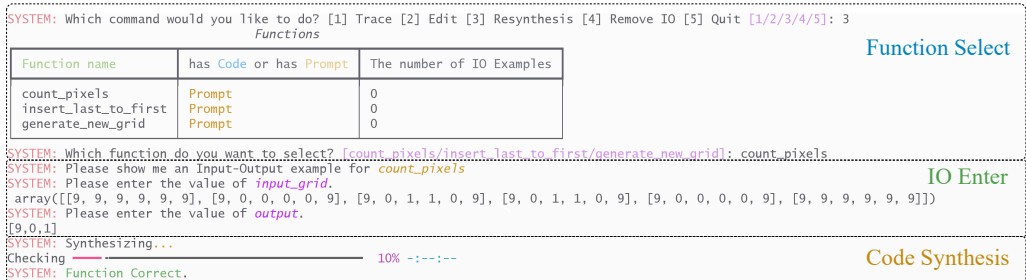

Figure 16: The resynthesis operation. Users can provide IO constraints and ask the compiler to resynthesize the program.

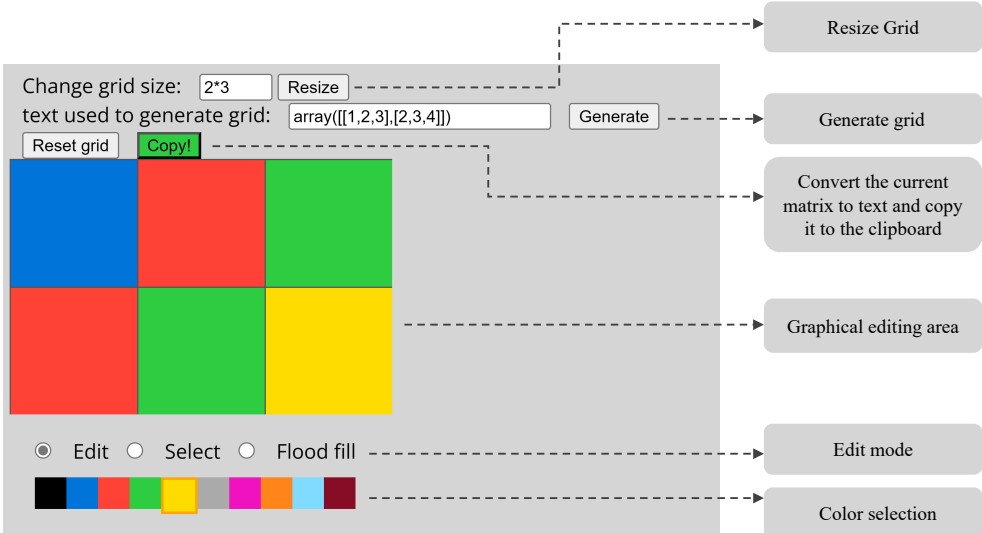

Figure 17: The grid editor.

## E.4 Suggested Prompts for System B

Users can input natural language without restrictions in system B. To enhance the user experience, we offer them a suggested prompt template shown in Figure 18.

## E.5 Task Assignment

Table 1 provides an assignment for ARC tasks. Each number corresponds to a question; a number highlighted in blue instructs use of System A before System B, whereas an orange number, conversely, denotes System B to be operated before System A.

```
You are a skilled Python programmer.
Your task is to write Python code to transform the input grid into the output grid.
In the input grid, you should see ...
To make the output grid, you should ...
Return your Python code in Markdown format.

```python
import numpy as np
black, blue, red, green, yellow, grey, pink, orange, teal, maroon = range(10)
def main(input_grid: np.ndarray) -> np.ndarray:
```
```

Figure 18: prompt for System B.

| User id | Tasks |
|---------|-------|
| 0 | 4 34 36 48 50 116 128 222 230 309 384 |
| 1 | 6 74 101 102 104 139 141 150 152 218 235 247 257 279 289 312 332 340 343 356 368 370 374 382 |
| 2 | 21 55 61 65 69 86 92 164 165 182 183 186 227 234 258 261 262 271 307 310 344 378 379 386 |
| 3 | 0 8 22 75 95 108 125 140 157 192 198 202 205 228 237 245 255 265 280 316 339 346 363 365 371 381 |
| 4 | 24 28 37 68 71 80 87 98 114 124 166 170 200 206 208 241 272 297 298 301 302 303 315 317 319 327 336 360 361 389 398 |
| 5 | 16 25 30 56 62 70 111 120 130 142 151 173 180 185 212 215 226 275 295 347 364 388 393 397 |
| 6 | 53 63 105 135 137 159 176 224 244 260 278 300 320 341 359 |
| 7 | 10 12 14 35 43 44 90 97 144 161 162 175 199 219 231 236 263 287 299 323 331 345 |
| 8 | 91 94 112 126 129 168 181 190 191 193 248 264 285 306 387 391 |
| 9 | 18 26 41 77 79 122 127 131 160 223 252 277 292 321 338 |
| 10 | 1 2 9 38 42 54 81 88 96 99 115 136 145 147 153 246 283 308 322 355 380 383 385 |
| 11 | 13 23 31 32 49 57 118 132 138 149 211 225 240 251 267 270 286 304 313 314 318 353 396 |
| 12 | 15 58 64 83 117 119 155 156 167 189 210 214 221 233 242 254 256 276 281 293 296 305 325 354 367 369 376 392 395 |
| 13 | 5 27 33 51 82 123 134 172 177 178 201 216 229 232 243 282 288 311 330 358 377 399 |
| 14 | 20 40 47 73 171 187 195 217 220 328 337 351 |
| 15 | 11 29 46 60 93 106 107 110 113 163 184 209 213 239 274 326 333 342 372 |
| 16 | 39 59 67 76 78 103 158 179 188 204 207 253 266 294 334 335 348 350 357 362 366 390 394 |
| 17 | 3 17 109 133 197 259 |
| 18 | 7 19 45 52 66 72 84 85 89 100 121 143 146 148 154 169 174 194 196 203 238 249 250 268 269 273 284 290 291 324 329 349 352 373 375 |

Table 1: Task assignment. Blue: use System A and then System B. Orange: use System B and then System A

# F   Additional Experiments and Analysis

In this section, we discuss the reasons for choosing ARC as our main benchmark, conduct further analysis of experimental results, and provide additional results on other tasks to comprehensively demonstrate the advantages and capabilities of ANPL.

## F.1   Benchmarks

**Why do we choose ARC instead of typical code generation benchmarks like HumanEval [9] and APPS [26].**    We evaluate ANPL on ARC instead of common code generation benchmarks [3, 9, 41, 44] according to the following three reasons: (1) We clarify that what we are measuring is the system's ability to convey the user's intent through interaction, not the system's ability to solve problems on its own (e.g. the user has a correct algorithmic solution in mind, but it is tedious to realize it in Python). In this regard, ARC's features (easy for humans to understand and solve but difficult to express via programming) would be more suitable than typical programming datasets such as MBPP and HumanEval. (2) We find that typical programming datasets are simpler for programming compared with ARC. The average line of code of MBPP, HumanEval, APPS, and ARC are 6.71, 7.78, 19.95, and 26.47 respectively. Competition datasets like APPS challenge the users to solve the tasks rather than generate code that fulfills users' intents. (3) Solutions for common code generation benchmarks can be found on the Internet, which runs the risk of data contamination. On the other hand, it is unlikely programmatic (python) solutions for ARC exist in any online corpus, making tasks in ARC unique enough for LLMs and participants of human study. However, to fully present the advantage and capability of ANPL, we have supplemented additional experiments with human studies on APPS benchmark, case studies on "realistic and long" programs, and automatic code generation on HumanEval benchmark.

**Why does the standard of "success" the accuracy of the training I/Os.**    Our experiment assesses the system's capacity to fulfill user intents rather than users' ability to design generalizable algorithms. User intent is defined as writing a program to pass the provided I/Os. Thus, we present the solving rate of test I/O in the main paper. Although it is a simplified setting, it is appropriate and sufficient taking into account the 440 man-hours invested already. However, we understand that readers may be interested in other information related to ARC itself, such as whether the Python code can be used to generate I/Os or whether DARC can provide insights into human decomposition strategies. Thus, we included the accuracy of all I/Os in the appendix as an addition.

## F.2   Additional Analysis on ARC

**The importance of the *control structure*s and *hole*s.**    In order to analyze the significance of the programming model, we conduct an analysis on the proportion of *control structure*s (*e.g.*, for, while, if) and *hole*s in programs that were correct in both system A and system B, and the proportion of programs that were correct in system A but incorrect in system B. In the programs that were correct in both systems, 47.8% utilized control flow, maintaining an average of 2.47 *hole*s. On the other hand, a remarkable 97.4% of programs that were correctly functioning in system A but failed in system B incorporated control flow, and had an increased average of 3.37 *hole*s. Results show that the introduction of *control structure* and *hole* has a significant positive impact on the user's ability to accomplish highly complex tasks.

**The distribution of time and the number of interactions.**    Furthermore, we conduct an analysis of the time consumption and the number of interactions involved in solved problems. That is, we filter the problems that can be solved by both systems A and B, collect the time and number of interactions spent by users, and then construct the two distributions shown in Figure 19. Results show that, for these tasks, system A exhibits faster completion times and fewer interaction counts (without TC) compared to system B, with slight advantages in terms of time and interaction. Nevertheless, this advantage is not highly pronounced, which could be attributed to the relatively low difficulty level of the problems that both system A and system B are capable of solving.

**Generalization ability.**    We further examine the accuracy of programs from System A and System B using all IO cases, see Table 2. System A and system B have similar generalization abilities based on the observation that for tasks passed on test cases by both two systems (224), the number of tasks

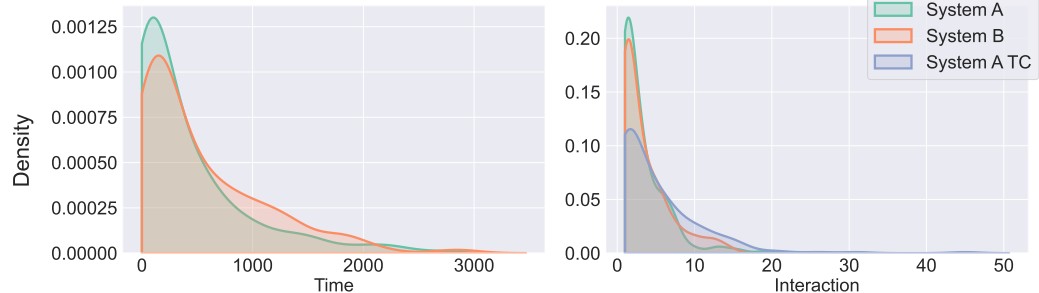

Figure 19: The distribution of time and the number of interactions. Trace Calculated (TC) means the trace mode is considered into interactions.

| A \ B | ✓✓ | ✓✗ | ✗✗ | Total |
|---|---|---|---|---|
| ✓✓ | 177 | 5 | 45 | 227 |
| ✓✗ | 10 | 32 | 31 | 73 |
| ✗✗ | 7 | 3 | 90 | 100 |
| Total | 194 | 40 | 166 | 400 |

Table 2: Generalization on ARC. ✓✓means the compiled Python program passes both train cases and test cases. ✓✗means the Python program passes the test cases but fails train cases. ✗✗means the program fails in all the cases.

that system A failed to generalize on all IO cases is 10/224 and the number of system B is 5/224, and these two numbers have no significant differences. Thus, we conclude that some programs failed to generalize to all IO cases because of the difficulty of the corresponding tasks and the designing of the underlying algorithm, which is independent of the usage of systems.

**GPT-4 results on System C and D.** During the experiment, we didn't have access to the GPT-4 API and it is impossible to conduct another large-scale human study for GPT-4 now. However, we can measure the solving rate w/o the interaction of ANPL+GPT-4 (System C) and GPT-4 (System D) with existing data: From the result in Table 3, we can conclude that (1) although a more advanced LLM would achieve higher accuracy, the utilization of ANPL always results in an improvement. (2) ARC still remains difficult for GPT-4 to give a 1-shot answer, making intent clarification a persistent challenge in using LLMs as programming assistants.

### F.3 Human Studies on APPS

Here we aim to discuss the generalization ability of ANPL on typical programming tasks and compare ANPL on programming ability against Parsel (not automatic code generation ability).

We conduct a small study with 4 participants and 20 randomly selected APPS-interview problems. Other domains considered were MBPP, HumanEval, and APPS-competition problems. We did not choose MBPP and HumanEval because they are much simpler compared with ARC, with many questions solvable using just a few lines of Python code (The [min, max, mean] of lines of ARC's (passed) program is [4, 113, 26.47] while MBPP is [2, 50, 6.71], HumanEval is [2, 32, 7.78]). Also, we did not test on competition-level tasks limited by user ability. Each participant was instructed to solve each problem with three systems (ANPL, GPT-3.5-turbo, and Parsel) appearing in a random order for a total of 30 minutes per system. We replicated Parsel with GPT-3.5-turbo because Codex is no longer available. As shown in Figure 20: (1) ANPL outperforms the original GPT-3.5-turbo and Parsel in final solving rate and solving time which is consistent with the result on ARC. Specifically, ANPL achieves 100% solving rate, GPT-3.5-turbo achieves 85%, and Parsel achieves 55%. Also, the 1-shot performance (without interaction) of them is 30%, 0%, and 15% respectively. (2) The solving rate for APPS-interview is higher than the one on ARC, indicating APPS-interview is easier than ARC for human interaction. In fact, the challenge of APPS lies in how users solve the tasks

Table 3: GPT-4 and GPT-3.5-turbo results on System C and D

| LLM | System C (ours w/o inter.) | System D |
|---|---|---|
| GPT-3.5-turbo | **23.50%** | 16.75% |
| GPT-4 | **30.50%** | 23.50% |

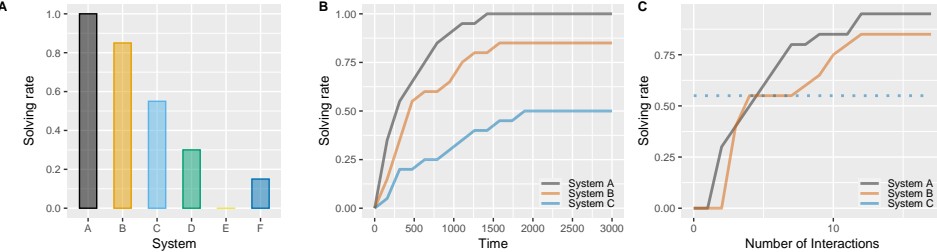

Figure 20: Results of APPS-interview. A: Solving rate of four systems, where System A is ANPL, System B is GPT-3.5-turbo, System C is Parsel, System D is ANPL w/o interaction, System E is GPT-3.5-turbo w/o interaction, System D is Parsel w/o modifying. Note that Systems C and D here are different from the main paper. Parsel is replicated with GPT-3.5-turbo and set to sample 10 Python programs each time, which has the sampling number as the resynthesis operation of ANPL. B: The relationship between solving rate and time consumption. C: The relationship between solving rate and number of interactions. Note that Parsel cannot effectively facilitate interaction so it is presented as a dashed line.

rather than grounding users' intents to code. (3) Parsel performs the worst because Parsel and ANPL have different design objectives: ANPL is designed to ground user intent to code through interaction while Parsel is designed for automatic reasoning without human intervention. Specifically, Parsel does not even retain the context of past interactions. This leaves the only method for "debugging" to continuously modify the initial Parsel code and provide more I/Os until it can be compiled to Python in a single attempt.

Overall, we conclude that (1) ANPL is generalizable to different domains and (2) ANPL and Parsel are substantially different systems: ANPL is designed as an interactive programming system to significantly reduce programming complexity and assure code quality, where the users can iteratively interact with the system to safely ground their minds to code; While Parsel is primarily a reasoning system, where the programming task is given to it "as is" and it leverages the capabilities of LLM to decompose and ground it to code without any user interventions.

## F.4 Case Studies on "Long and Realistic" Programming Tasks

To further show the ANPL's generalization ability, we conduct a qualitative study by using ANPL to compile several projects drawn from CS courses and GitHub: text editor, robot control, naive MAGIC card game, and LS-8 CPU. These tasks are the prototypes of realistic applications and are representative in terms of the program form with complex control flows and many sub-functions. They are much longer (198 lines of code on average) than existing program synthesis benchmarks (MBPP[6.71], HumanEval[7.78], APPS[19.95]).

It is shown in Figure 21 that ANPL can be used to implement these applications with much less coding effort (with 33.21% lines of code) and obviously lower programming barrier (by using natural language) than Python. Such results show that ANPL has the potential to be adopted in real scenarios with great superiority of lower programming complexity and better code quality assurance.

## F.5 Automatic Code Generation on HumanEval

**Settings.** Following Parsel [71], we generated 100 test cases with temperature=0.6 for each problem in HumanEval. Then we generated 4 high-level solutions with temperature=0.6 and corresponding ANPL codes with temperature=0.2 and presence_penalty=0.1. For each ANPL code, we generated 16 Python implementations for each function. Then we sampled the programs composed of generated

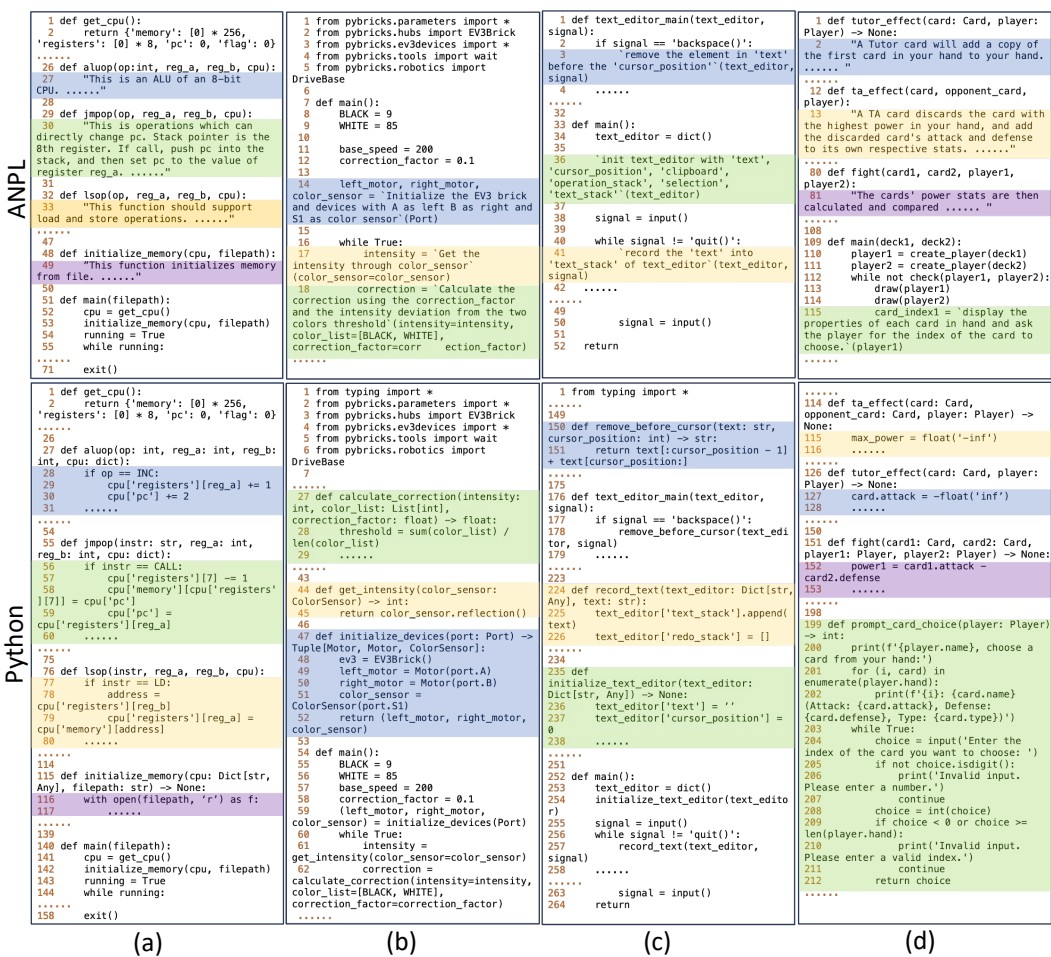

Figure 21: The ANPL and Python code of four projects: (a) LS-8 CPU (b) a robot controller (c) a text editor (d) a naive MAGIC. We present part of the code and mark the ANPL code and its corresponding Python code with the same background color.

functions and evaluated them by generated test cases. If one program passed all the test cases, it would be chosen as the final submission and evaluated by the real test cases. Otherwise, we would debug for each function of the program that passed the most generated test cases. In function-level debug, we used the execution traces to ask LLM for 8 fixed functions with temperature=0.6. Then we repeated the evaluation and debug processes for at most 2 rounds. For LLMs chosen, we use GPT-3.5-turbo-0301 and GPT-4-0314[3]. Our prompts are shown in Figure 22, 23, 24, 25.

**Results.** As shown in Table 4, we can conclude that although ANPL itself is not designed for automatic code generation tasks, due to its explicit recursive decomposition property, this framework has a certain advantage when used for automatic code generation. It achieved 76.22% on GPT-3.5 and 86.59% on GPT-4, both significantly surpassing the original LLM model. Furthermore, the design of ANPL for natural programming debugging was not reflected in this test, so there is still room for improvement in ANPL's performance in automatic code generation tasks, which we will consider as future work

---

[3]Note that we did not check the detailed date carefully and this might be different from the other's choices due to OpenAI API change

| Method | LLM | Pass@1 |
|---|---|---|
| Original | GPT-3.5 | 56.7 |
| CodeT [8, 53] | GPT-3.5 | 65.8 |
| Reflexion[53] | GPT-3.5 | 68.1 |
| ANPL (ours) | GPT-3.5 | **76.22** |
| Original [45] | GPT-4 | 67 |
| Original [7] | GPT-4 | 82 |
| Original [53] | GPT-4 | 80.1 |
| Reflexion [53] | GPT-4 | **91.0** |
| (Parsel + CodeT) [71] | GPT-4 | 85.1 |
| ANPL (ours) | GPT-4 | 86.59 |

Table 4: GPT-3.5 and GPT-4 Pass@1 accuracy on HumanEval [9].

```
-----Question-----
{question}
-----Task-----
Give an assert test for this question in the following format. Each assert statement should
  ↪ be on one line.

-----Test-----
```
assert
```

You should only output the assert test. Omit explanations or any additional text.
```
```

Figure 22: Test generation prompt for HumanEval.

```
-----Question-----
{question}

-----Task-----
Propose a clever and efficient high-level solution for this problem. Consider all edge cases
  ↪ and failure modes.

Some common strategies include:
    Constructive algorithms, Binary search, Depth-first search (DFS) and similar algorithms,
      ↪ Dynamic programming, Bitmasks, Brute force, Greedy algorithms, Graphs, Two pointers,
      ↪ Trees, Geometry, Graph matchings, Hashing, Probabilities, Data structures, Sortings,
      ↪ Games, Number theory, Combinatorics, Divide and conquer, Disjoint set union (DSU),
      ↪ Expression parsing

Let's think step by step to come up with a clever algorithm.
You should only output high-level solution without any pseudocode or code.
```

Figure 23: High-level solution prompt for HumanEval.

```
-----Question-----
{question}

-----Solution-----
{solution}

-----Task-----
Translate the solution for the problem into python code in the following format:

-----Program-----
```
python code
```

You should define some helper functions before function {func_name} to decompose it. Each
 ↪ function should be described by docstring.
You should only output the python code! Omit explanations or any additional text!
```

Figure 24: Translation to ANPL prompt for HumanEval.

```
-----Question-----
{question}

-----Solution-----
{solution}

Here is an program implementation of the solution.
-----Program-----
{program}

Here is a function of the program with input-output traces.
-----Function-----
{function_with_traces}

-----Task-----
If there are some mistakes or exceptions in the function, return the fixed function. You can
 ↪ define helper functions before it to decompose it into sub-functions.
Your output should be in the following format:
```
def {func_name}(...):
    '''
    The description of the function.
    '''
```
You should only output the function code! Omit explanations or any additional text!
```

Figure 25: Function-level debug prompt for HumanEval.

# G   DARC Details

| time | role | action | content |
|------|------|--------|---------|
| 1684141011.7825 | user | enter code | "def main(input):
    centers = \`traverse the input which is a 2-dim numpy array, return positions which satisfies that there is no grey in its 3*3 neighbor\`(input)
    scores = \`for each position in the centers, count the yellow position in its 3*3 neighbor\`(input, centers)
    max_score = np.max(scores)
    center_yellow = centers[scores==max_score]
    center_black = centers[scores!=max_score]
    output = \`for each position in the position list, make its 3*3 neighbor yellow\`(input, center_yellow)
    output = \`for each position in the position list, make its 3*3 neighbor black\`(output, center_black)
    return output" |
| 1684141011.7839 | system | parser | "user enter correct code" |

Figure 26: The header of CSV files.

```python
def main(input):
    centers = "traverse the input which is a 2-dim numpy array, return positions which satisfies that
    ↪  there is no grey in its 3*3 neighbor"(input)
    scores = "for each position in the centers, count the yellow position in its 3*3 neighbor"(input,
    ↪  centers)
    center_yellow, center_black = "return the center with the max scores and other centers"(centers,
    ↪  scores)
    output = "for each position in the position list, make its 3*3 neighbor yellow"(input, center_yellow)
    output = "for each position in the position list, make its 3*3 neighbor black"(output, center_black)
    return output
```

Figure 27: An ANPL program in DARC.

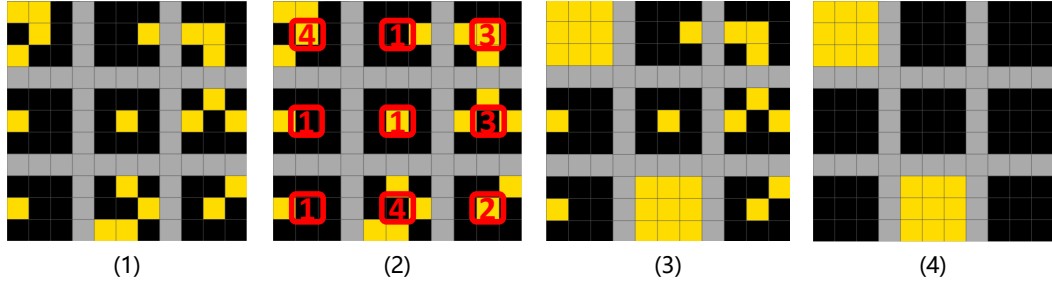

(1)          (2)          (3)          (4)

Figure 28: The trace of the given ANPL program. (2) Centers are framed by red rectangles and we mark scores inside each rectangle.

The Recursive Decomposition Dataset of ARC Tasks (DARC) is an assemblage of interaction records associated with 400 ARC tasks. These records involve the intercommunication between users, the system, and GPT-3.5-turbo. Figure 26 presents the header of each CSV file. For each interaction, data concerning the role, action, content, and timestamp are completely collected and stored. The result is the entire user interaction history with our systems can be perfectly *replayed* at a later time. However, the response from LLMs will be different for the following reasons: (1) we used temperature = 1.0, which will cause different tokens to be sampled (2) the GPT-3.5-turbo implementation might be changed, which is outside of our control.

```python
import numpy as np
from typing import *
(black, blue, red, green, yellow, grey, pink, orange, teal, maroon) = range(10)

def get_max_score_center(centers: List[Tuple[int, int]], scores: np.ndarray) -> Tuple[List[Tuple[int,
↪    int]], List[Tuple[int, int]]]:
    max_score = np.max(scores)
    max_centers = [centers[i] for i in range(len(centers)) if scores[i] == max_score]
    other_centers = [centers[i] for i in range(len(centers)) if scores[i] < max_score]
    return (max_centers, other_centers)

def find_positions_without_grey_neighbors(input: np.ndarray) -> List[Tuple[int, int]]:
    positions = []
    for i in range(1, input.shape[0]-1):
        for j in range(1, input.shape[1]-1):
            if np.all(input[i-1:i+2, j-1:j+2] != grey):
                positions.append((i, j))
    return positions

def make_neighbors_yellow(input: np.ndarray, positions: List[Tuple[int, int]]) -> np.ndarray:
    for position in positions:
        input[position[0]-1:position[0]+2, position[1]-1:position[1]+2] = yellow
    return input

def make_neighbors_black(input: np.ndarray, positions: List[Tuple[int, int]]) -> np.ndarray:
    for position in positions:
        input[position[0] - 1:position[0] + 2, position[1] - 1:position[1] + 2] = black
    return input

def count_yellow_neighbors(input: np.ndarray, centers: List[Tuple[int, int]]) -> np.ndarray:
    scores = np.zeros(len(centers))
    for i, position in enumerate(centers):
        scores[i] = np.sum(input[position[0] - 1:position[0] + 2, position[1] - 1:position[1] + 2] ==
        ↪    yellow)
    return scores

def main(input):
    centers = find_positions_without_grey_neighbors(input)
    scores = count_yellow_neighbors(input, centers)
    center_yellow, center_black = get_max_score_center(centers, scores)
    output = make_neighbors_yellow(input, center_yellow)
    output = make_neighbors_black(output, center_black)
    return output
```

Figure 29: The compiled Python program in DARC.

300 of the 400 ARC tasks were effectively decomposed and converted into Python using the ANPL, averaging 2.7 *hole*s per task – these tasks solves the test-input, but some may fail to generalize to other input-outputs, see Appendix F.2. The DARC dataset not only houses the final solutions to the ARC tasks but also encapsulates the diverse problem-solving approaches employed by different users. Crucially, it is a dataset detailing how humans decompose abstract procedural tasks into simpler sub-tasks, and ground each task into a program (e.g. Python) in collaboration with an LLM. We give an example of an ANPL program, the compiled Python program and its corresponding trace in Figure 27, Figure 29 and Figure 28.

The DARC dataset provides a valuable window into the system's task completion processes. By documenting ANPL decompositions, Python code, and detailed interaction histories, it permits us to gain insights into the practical application of Language Learning Models (LLMs) for programming. We hope that this dataset will be useful for others seeking to understand and refine similar systems.

## H Computational Resources

For our human study, LLM APIs were called 4304 times for System A (10.76 per task), and 1923 times for System B (4.81 per task). The distributions of the number of LLM API calls are presented in Figure 30. Since System A generates 10 candidates when resynthesizing, the number of API calls on several tasks exceeds forty times. For most tasks, the number of interactions in the two systems is less than twenty.

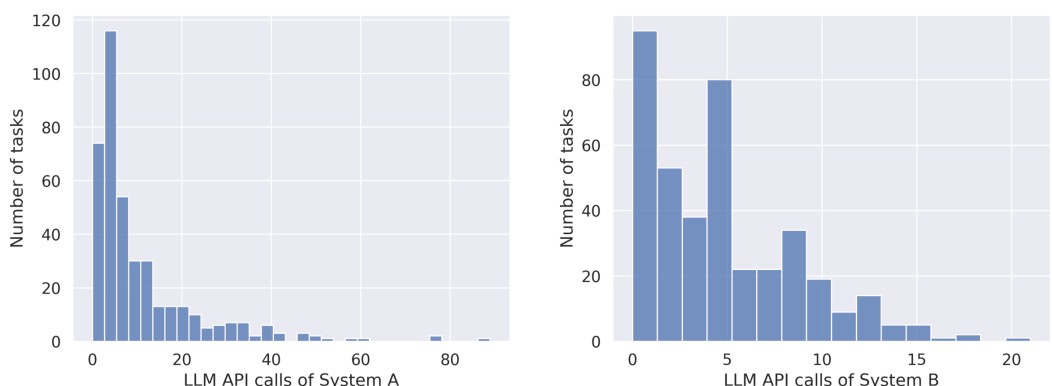

Figure 30: LLM API calls of Systems A and B.

# I   Case Study

**A case of user interaction.**   Figure 31 serves as an illustrative instance. Initially, the user enters the ANPL code into the system, which consequently produces a function for every hole and automatically verifies the code's validity. Upon encountering a programming error, the system issues a warning to the user. In response, the user examines the input and output of the *find_smallest_unit* function and discovers it doesn't align with expectations. The user then adjusts the natural language description, leading to ANPL successfully meeting the test input and output samples following a code regeneration. Eventually, the system presents the complete code to the user, as shown in the Figure 32.

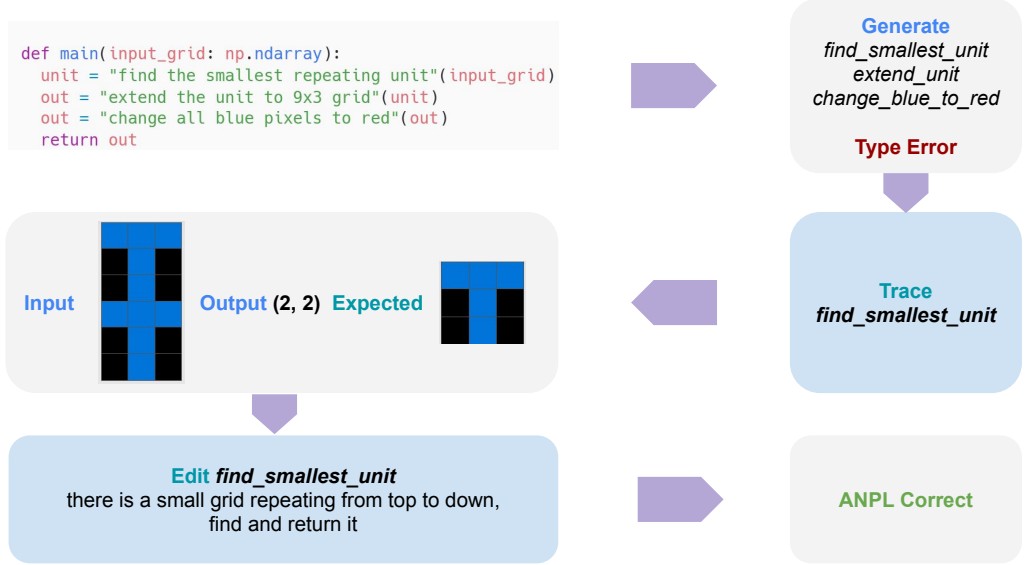

Figure 31: User Interaction with System A and Subsequent Code Generation Process

**Difficult Tasks.**   From the overall 400 tasks under consideration, several tasks pose a significant challenge to users because it is hard or impossible to find the solving logic. We present one case in Figure 33.

**User-Specific Solution.**   The potential solution of the ARC problem is heavily reliant on the user's algorithm design, which has an impact on the difficulty of programming. For instance, in the ARC task illustrated in Figure 34, some users might attempt to identify the color pattern across each row or column. Others might note the black square's location, then rotate the grid 90 degrees and select the grid in the same position as the answer. The first approach is quite challenging, often too demanding to be completed within a time constraint. Conversely, the latter method leverages the rotational symmetry of the grid, enabling a direct translation into the correct Python code, eliminating the need for debugging.

```python
import numpy as np
from typing import *
(black, blue, red, green, yellow, grey, pink, orange, teal, maroon) = range(10)

def find_smallest_unit(input_grid: np.ndarray) -> np.ndarray:
    """
    Given a 2D numpy array, this function finds the smallest repeating unit in the grid and returns it.
    """
    (width, height) = input_grid.shape
    for i in range(1, width+1):
        if width % i == 0:
            for j in range(1, height+1):
                if height % j == 0:
                    unit = input_grid[:i, :j]
                    if np.all(np.tile(unit, (width//i, height//j))[:width, :height] == input_grid):
                        return unit
    return input_grid

def extend_unit(unit: np.ndarray) -> np.ndarray:
    """
    Given a 2D numpy array, this function extends the array to a 9x3 grid.
    """
    (width, height) = unit.shape
    if width >= 9 and height >= 3:
        return unit[:9, :3]
    else:
        extended_unit = np.zeros((9, 3), dtype=unit.dtype)
        for i in range(9):
            for j in range(3):
                extended_unit[i, j] = unit[i % width, j % height]
        return extended_unit

def change_blue_to_red(unit: np.ndarray) -> np.ndarray:
    """
    Given a 2D numpy array, this function changes all blue pixels to red.
    """
    unit[unit == blue] = red
    return unit

def main(input_grid: np.ndarray):
  unit = find_smallest_unit(input_grid)
  out = extend_unit(unit)
  out = change_blue_to_red(out)
  return out
```

Figure 32: The code synthesized by ANPL.

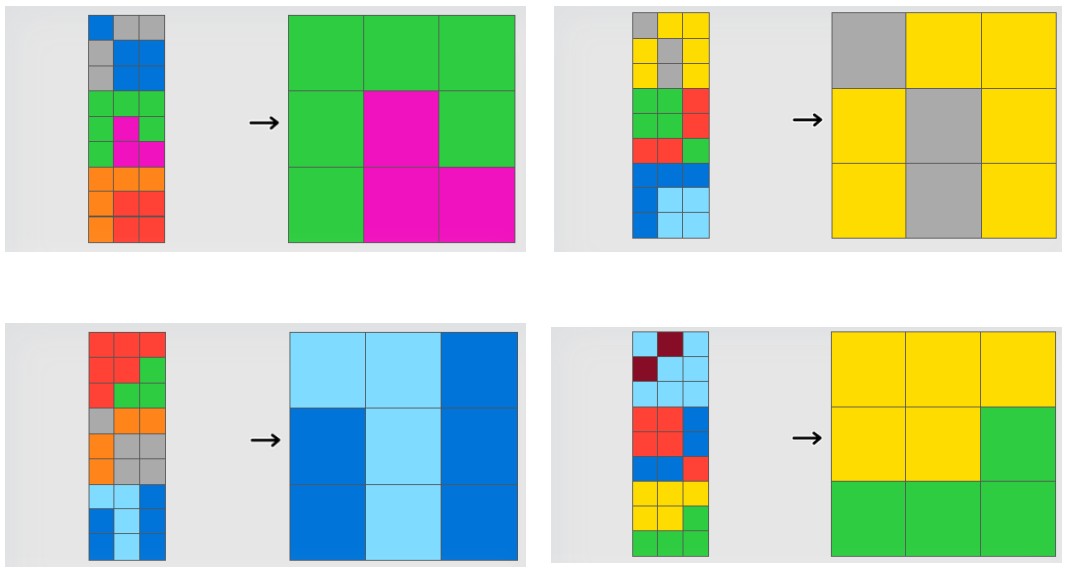

Figure 33: Unsolvable ARC task for the user.

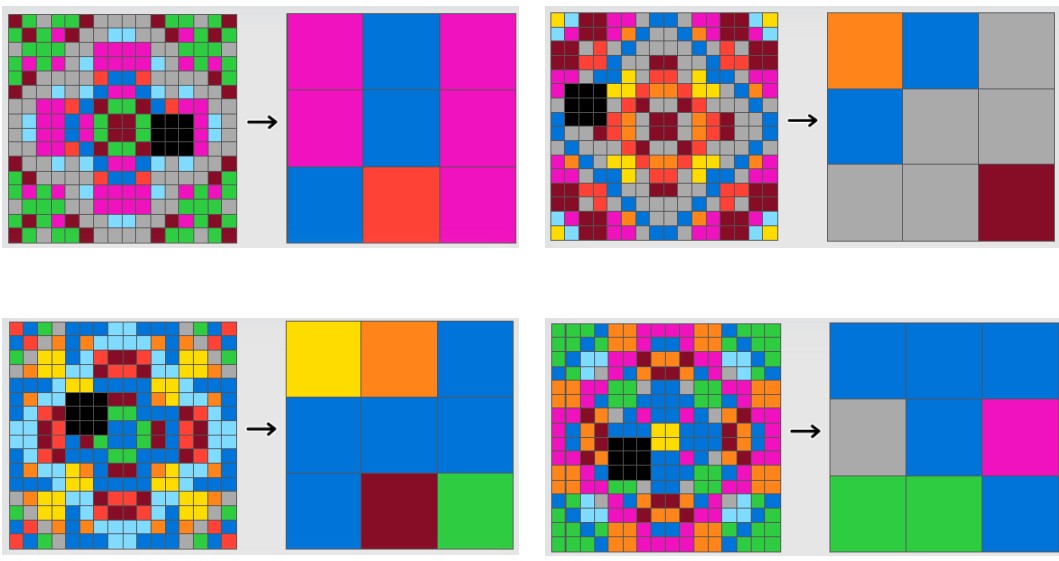

Figure 34: Example of an ARC task demonstrating differing user strategies.

