# OpenReview forum: "ANPL: Towards Natural Programming with Interactive Decomposition"
_NeurIPS.cc/2023/Conference — NeurIPS 2023 poster_

### Official Review · Reviewer_ecZq · 2023-07-05

**Soundness:** 3 good
**Presentation:** 2 fair
**Contribution:** 2 fair
**Rating:** 7
**Confidence:** 4

**Summary:**

In this paper, the authors introduce Abstracted Natural Programming Language (ANPL), a programming language and system to improve Large Language Model (LLM) assisted programming. ANPL programs are structured in a top-down manner; a top level program called the "sketch" solves the task by specifying the data-flow between multiple secondary level sub-functions called "holes". Each "hole" encapsulates a textual description of its targeted functionality. This ANPL program is then translated into a conventional python program via the ANPL compiler, which uses LLMs to implement the "holes".  Further, to help debug the generated/compiled program, the system provides features such as exposing execution traces and intelligently merging novel user edits to hole descriptions. Human study shows that compared to using LLMs naively, ANPL helps users solve programming tasks faster and with higher success rate. Finally, the paper also contributes a dataset of textual decomposition (by humans) of 300/400 reasoning tasks in the ARC benchmark, which can provide insights into how humans decompose programming tasks.

**Strengths:**

### Originality
The idea of improving programming by decomposing the program into sketches and holes in itself is not a novel idea (the authors properly cite previous works along this direction). However, structuring a LLM-augmented programming interface around this system is indeed novel.

### Quality
The paper is well written. Further, it presents insightful experiments and ablations in the main paper as well as the extended appendix.

### Significance
This paper targets the fastly-evolving direction of LLM assisted programming, and hence will be of interest to the research community at large. While "sketch-hole decompostion" is commonly used in LLM assisted programming, this paper provides a thorough human study to firmly establish the benefits of such an approach.


**Weaknesses:**

While overall I believe its a well written paper, there are certain weaknesses with the presentation and the experiments section.

### Presentation / Clarity

1) The main paper does not clearly show *what* an ANPL program looks like - what is the input to the LLM?. The main paper only refers to Figure 1 which shows an ANPL program pictorially. While the ANPL compiler (written in ANPL) is in the paper, I believe it would be easier to understand the system with a clearer and explicit presentation of what a "sketch" is and what a "hole" is programmatically.

2) The main paper does not discuss the actual interface that the user uses (only shown in the appendix). As this is a crucial contribution of a "programming" system, I think it should also be shown in the main paper.

3) In general, the paper entangles a methodology contribution (using a sketch-hole paradigm for LLM assisted programming) with a system contribution (The interface, debugging features, merging features, test-case creation tools).

### Experiments

1) The interface seem to have additional features such as grid editor to create test cases etc. which may contribute to the user success. This make it harder the discern what parts of ANPL are essential.

2) An important baseline missing is users directly programming in python. On roughly 30% of the tasks, even with ANPL users take ~15 mins. Looking at the ARC challenge examples, I think directly programming in python may be faster than programming with ANPL (or the other baselines).

3) Program solutions which satisfy test cases but fail for train cases are counted as "successful" while measuring the success rate.  (Refer Table 2 appendix and results in the main paper).  If we count only the cases where both test and train cases are successful, the performance drops to 56.7% for ANPL versus 48.5% for naive LLM usage, which is significantly lower than the numbers used in the main paper. I think that successful programs should satisfy both train and test cases.

### Others

1) Many real world programming tasks are unlike the problems in ARC. Hence, its unclear if ANPL would be better than naive LLM systems in typical programming use-cases as well.

2) Another drawback of using ARC is that many tasks may involve sub-functions which are similar. Therefore users may benefit from know "what prompts works better" for certain sub-functions, and use them directly in future prompts (sort of saving the previously discovered functions). Therefore, a task benchmark where the tasks were more diverse might have been more suitable.

3) As GPT-4 seems to be much more performant than GPT-3.5 (used in the experiments) I wonder if the gap between ANPL and naive LLM usage is further decreased when larger LLMs are used.





**Questions:**

I thank the authors for providing an extensive appendix which answers many of the questions I had. However, a few more are as follows:

1) How often do users get the "sketch" wrong - are there some cases where the failure is due to wrong sketch while the hole implementations are correct?

2) Can the authors briefly talk about the learning curve of using ANPL?

3) Can the authors justify the choice for using train-fail-test-success as successful programs while measuring success rate?




**Limitations:**

The authors have adequately addressed the limitations and potential negative societal impact of their work.

---

> ### Author Rebuttal · Authors · 2023-08-09
>
> Thanks for your review comments! We provide detailed responses to all of your concerns. We'll revise our paper to make your concerns clear. We have a space limit so please let us know of any additional comments.
> ## Weaknesses
> ### Presentation
> Thanks for the suggestions and we'll revise the paper.
> However, we want to clarify that our main contribution is clear: designing the programming system to address the inability of existing LLMs that hardly clarify a user's intent precisely when a misunderstanding arises. Techniques like sketch-hole, merging, etc. are key aspects of making such a system.
> ### Experiments
> > 1.The interface seem to have additional features such as grid editor... make it harder the discern what parts of ANPL are essential.
>
> Sorry for the misunderstanding. The grid editor is provided for both System A and B. It is just a tool to improve the user experience. The comparison between the two systems is absolutely fair.
> > 2.... directly programming may be faster than programming with ANPL.
>
> In fact, Python and ANPL are fully compatible as mentioned in Sec. 3. The reason for the time cost on simple tasks is that during programming, much time is spent waiting for GPT's response. The average waiting time is 14.6% for System A and 23.5% for System B. Besides, the users were instructed to solve the tasks as fast as possible and they choose not to program directly when using System A because it would require more effort.
>
> It is impossible to recruit the same users for Python experiments, but we can demonstrate the user effort in ANPL and Python as the lines of code: the [min, max, mean] of lines of ANPL is [2, 76, 10.29] while the one of Python is [4, 113, 26.47], and note that writing natural language is simpler than coding.
> We also demo the ANPL's application on projects including MAGIC card game and LS-8 CPU with 67% lines of code reduction, see Global Rebuttal (3).
> > 3.The standard of "successful"...
>
> Our experiment assesses the system's capacity to fulfill user intents rather than users' ability to design generalizable algorithms. User intent is defined as writing a program to pass the provided I/Os. Thus, we present the solving rate of test I/O in the main paper. Although it is a simplified setting, it is appropriate and sufficient taking into account the 440 man-hours invested already.
>
> However, we understand that readers may be interested in other information related to ARC itself, such as whether the Python code can be used to generate I/Os or whether DARC can provide insights into human decomposition strategies. Thus, we included the accuracy of all I/Os in the appendix as an addition.
> ### Others
> > 1....if ANPL would be better in typical programming use-case.
>
> We have conducted additional experiments on the APPS dataset and implemented several representative long programming tasks to show ANPL's generalization ability. The results show that ANPL outperforms baselines consistently. Please refer to Global Rebuttal (1) (2) (3) for more details.
>
> In addition, using ARC as a programming task is actually appropriate.
> What we are measuring is the system's ability to convey the user's intent through interaction, not the system's ability to solve problems on its own. In this regard, ARC's features, which are easy for humans to understand and solve but difficult to express via programming, would be suitable. Refer to Global Rebuttal (4) for more details.
> > 2.... the drawback of ARC is users may benefit from knowing "what prompts work better" for certain sub-functions...
>
> As mentioned above, we have provided additional experiments on the APPS dataset to show ANPL's advantage (Global Rebuttal (1)).
>
> However, this concern might be a misunderstanding because
> (a) ARC already encompasses a large diversity of concepts based on its design principle [1] and is more complicated than datasets such as HumanEval and MBPP (Global Rebuttal (4)). In fact, even competition datasets involve repetitive algorithmic codes, such as BFS, DFS, DP, etc.
> (b) Users can do exactly the same thing with GPT so it won't have any impact on the comparison between the two methods.
>
> Actually, this is a very interesting problem i.e. "library learning with ANPL/prompt", which is discussed in the paper's future work.
>
> [1] On the Measure of Intelligence
> > 3.... if the gap between ANPL and naive LLM usage is further decreased when larger LLMs are used.
>
> During the experiment, we didn't have access to the GPT-4 API and it is impossible to conduct another large-scale human study for GPT-4 now. However, we can measure the solving rate w/o the interaction of ANPL+GPT-4 (System C) and GPT-4 (System D) with existing data:
> |LLM|System C|System D|
> |-|-|-|
> |GPT-3.5-turbo|23.5%|16.75%|
> |GPT-4| 30.5%|23.5%|
>
> We can conclude that (a) although a more advanced LLM would achieve higher accuracy, the utilization of ANPL always results in an improvement.
> (b) ARC still remains difficult for GPT-4 to give a 1-shot answer, making intent clarification a persistent challenge in using LLMs as programming assistants.
>
> ## Questions
> > 1.How often do users get the "sketch" wrong...
>
> Determining when a situation belongs to "writing a wrong sketch" is difficult. Instead, we quantify the frequency of users modifying the main function, which encompasses cases of writing a wrong sketch. We find that in 45.67% of successful tasks, users have modified the main function but finally they got the right code, showing that users often make mistakes when coding but the mistakes can usually be corrected through interaction.
> > 2.... learning curve of using ANPL?
>
> The learning curve of ANPL is shown below.
> |time stamp|1|2|3|4|5|6|7|
> |-|-|-|-|-|-|-|-|
> |solving rate|80.11%|77.30%|77.22%|84.58%|71.13%|57.78%|70.83%|
>
> We conclude that there is no significant learning curve. This might be due to substantial variations in difficulty across different tasks.
> > 3....why using train-fail-test-success as success?
>
> Please refer to Weaknesses-experiments-3 provided above.

---

> > ### Comment · Reviewer_ecZq · 2023-08-16
> > **Thank you for the rebuttal!**
> >
> > ## Summary
> > Thank you for the detailed rebuttal. The rebuttal, along with the other reviews has helped improve my understanding of the proposed system. I have decided to increase the score of the paper to 7-accept as the authors have addressed almost all of the weaknesses and questions I raised.
> >
> > ## Response to the rebuttal in detail
> >
> > 1) Presentation - It is unclear what are the changes the authors are planning. How are the authors planning to revise the paper? My suggestion would to at least include one figure of the interface. Further, it would be useful if figure one explicitly shows the "sketch" code as well as the "hole" code (instead of a pictorial representation). I would be happy to hear the author's view.
> >
> > 2) Experiments:  Thank you for clarifying that the grid is provided for all the ablated systems. I find it interesting that a lot of the time is spent waiting for the GPT prompts. I wonder if the users were told to use python functions that they code up if necessary. Nonetheless, the lack of a python only baseline is but a minor drawback. My third query, on test-set vs train-set scores is also well addressed. I recommend that the authors include the stated reasons in the appendix while discussing these statistics.
> >
> > 3) Others - I appreciate the results in the real-world program setup, it addresses my first two issues. I thank the authors for results with GPT-4. Since ANPL's goal is reasoning about user intent (rather than solving a task), better models would perhaps improve ANPL as well.
> >
> > Also, thank you for answering my questions! I find the results useful to improve my understanding of ANPL.

---

> > > ### Author Response · Authors · 2023-08-16
> > > **Thanks for your reply!**
> > >
> > > Thanks for the reply and increasing the score! We are glad that our rebuttal has addressed most of your concerns. Also, thanks for your effort in reading our paper and other reviews thoroughly!  It is our pleasure to have a further discussion with you.
> > >
> > > ## Presentation
> > > Good suggestions! Sorry for our ambiguous response before. We are limited to 6,000 characters in our rebuttal.
> > >
> > > **Following your advice**, we promise that (1) we will add an interface figure in the "Human Studies" section of the main paper (we do not want to add it in the "method" section because the core contribution of ANPL is its programming paradigm and its compiler instead of the interface, but we will add an ANPL code example in the "method" section, see below). (2) We will add an ANPL code example in the main paper.
> > >
> > > However, it might not be appropriate to modify the ANPL example in Figure 1 from pictorial representation to code, as we initially attempted this, and it could make the example appear complex, failing to highlight the concept of separating control/data flow and functions in ANPL. We will try our best to modify Figure 1 and if we failed, we will add a new figure to show an ANPL code example. (more suggestions are welcome)
> > >
> > > ## Experiments
> > > We did not explicitly forbid the users to write Python code in our experiment so **they can write Python code if they want**. We count the number of pure Python functions written by users and find that only 5 functions are written in pure Python (1261 functions in total), with an average length of 7.2 lines (including `def` and `return`). This indicates that users can choose to write Python programs, but most of the time, **for the sake of efficiency, they choose to write ANPL**.
> > >
> > > Although we said that ANPL is compatible with Python and that "the users were instructed to solve the tasks as fast as possible and they choose not to program directly when using System A because it would require more effort", **we agree with you that a pure Python baseline should be added to make this result clearer. But we cannot afford to do it**. In fact, we are constrained by the complexity of the human study, and each additional baseline would significantly increase the cost of the human study. For instance, we have already completed a 440 man-hours human study, which is currently the most comprehensive experiment on users known about the LLM. The difficulty of conducting this experiment might be much greater than what readers may imagine. If we were to add a Python baseline, the cost of this experiment would extend to 660 man-hours. This not only demands more time but also incurs higher costs of money and manpower (few people want to finish an ARC problem 3 times). We believe that the existing experimental results and the above analysis (line of code and the number of pure Python functions) **have adequately demonstrated the advantage of ANPL**.
> > >
> > > ## Others
> > > Yes, the performance of ANPL will be improved as LLM upgrades and we will revise our paper to add this GPT-4 result. As you have seen, the purpose of ANPL is to construct a programming assistant capable of accurately conveying user intent through interaction (like debugging) instead of an automatic problem-solving machine. Increasing LLM's capabilities will make this interaction easier, but the gap of understanding between the user's intent and the LLM's interpretation will always be there.
> > > Even if the LLM surpasses programmers in the future, it is still necessary for users to communicate with LLM to clarify user intent -- where ANPL comes in.
> > >
> > > ## Conclusion
> > > In summary, we plan to
> > >
> > > (1) add an interface figure in the "Human Studies" section of the main paper
> > >
> > > (2) add an ANPL code example in the main paper
> > >
> > > (3) clarify that the grid editor is provided for all the ablated systems in the appendix
> > >
> > > (4) provide the GPT response time analysis in the experiment part
> > >
> > > (5) make the experimental setup clearer like whether the users can write Python functions in the appendix
> > >
> > > (6) provide the comparison between lines of ANPL code and Python code in the appendix to show ANPL's advantage compared with Python
> > >
> > > (7) clarify the choice of test-set vs train-set in the appendix
> > >
> > > (8) add GPT-4 results (w/o interaction) in the appendix
> > >
> > > (9) add the results on APPS and "long and realistic" programming tasks (Global Rebuttal) in the appendix
> > >
> > > Again, we greatly appreciate your thorough review of our paper, and we thank you for your suggestions for improving the quality of the paper! If you have any questions, please feel free to post them and discuss them with us. We firmly believe that ANPL and similar frameworks will undoubtedly play a significant role in the area of AI assistants.

---

> > > > ### Comment · Reviewer_ecZq · 2023-08-16
> > > > **Thank you for the reply!**
> > > >
> > > > The planned updates to the presentation seem reasonable. Thank you for the details on my queries regarding "experiments" and "others".
> > > > I believe the paper will benefit from the planned updates and hopefully further propels research in LLM + human-in-the-loop systems.

---

> > > > > ### Author Response · Authors · 2023-08-16
> > > > > **Thanks for the fast reply!**
> > > > >
> > > > > Thanks for your reply and effort in improving the paper! We'll revise our paper carefully.

---

### Official Review · Reviewer_tgZy · 2023-07-06

**Soundness:** 4 excellent
**Presentation:** 4 excellent
**Contribution:** 2 fair
**Rating:** 5
**Confidence:** 3

**Summary:**

This paper proposes ANPL, a programming system for user interaction with an LLM for code generation. The main idea is to have the user generate high-level dataflow and program structure, while directing the LLM to fill in the "holes." They present results of a user study of ANPL, where users are asked to write programs to solve ARC tasks, and show that ANPL outperforms a baseline of (undirected) user interaction with GPT-3.5 via ChatGPT, and one-shot program synthesis using ANPL and GPT-3.5.

**Strengths:**

This paper presents a novel method of incorporating LLMs into a programming workflow. A major strength is the design of ANPL. Though the idea of decoupling a synthesis problem into a hole and a sketch is not new in program synthesis (or even neural program synthesis [1]), the application (and execution) here for an LLM-driven coding assistant seems novel and natural.

The user study is also significant, and contains a lot of great analysis.

The writing is also very clear.

[1] Nye, Maxwell, et al. "Representing Partial Programs with Blended Abstract Semantics." International Conference on Learning Representations. 2020.

**Weaknesses:**

My major concern is that the main contributions are (1) a programming system and (2) its evaluation. There is not much novelty on LLMs or even neural program synthesis.

Some additional points about the design of the user study:
- Most users in the user study were unfamiliar with code generation using LLMs (Figure 5 in the appendix), and only 1 prompt was provided as example (Figure 10 in the appendix). Therefore, it's not clear how much the difference in performance is due to inexperienced users of System B. For instance, the design of the study could have provided both the ANPL prompt (Figure 4 in the appendix) to the users as well, or given them some basic training first.
- ARC is designed as a test of human intelligence, so the performance of a human + ANPL on ARC doesn't directly translate into performance on writing programs. Ideally the results would have included an additional dataset designed for synthesis (such as MBPP or HumanEval).

**Questions:**

Can you provide results for users aggregated by the language model code generation experience?

The performance of System B appears to saturate very quickly. This could be due to a degradation in performance of System B over time, for instance, as the context grows too long. Were the contexts for the LLMs reset at any point for either ANPL or System B?

**Limitations:**

Yes

---

> ### Author Rebuttal · Authors · 2023-08-09
>
> Thank you for your review! Your concerns mainly involve some questions about the experimental setup. In response to this, we have provided comprehensive explanations and conducted an additional experiment on APPS and complicated programs. The results show that ANPL still outperforms baselines and can generalize to broader domains. Please refer to the detailed responses below and the Global Rebuttal for more information.
>
> ## Weaknesses
>
> > Most users in the user study were unfamiliar with code generation using LLMs... Therefore, it's not clear how much the difference in performance is due to inexperienced users of System B. For instance, the design of the study could have provided the ANPL prompt (Figure 4 in the appendix) to the users as well, or given them some basic training first.
>
> Although some users are unfamiliar with System B, they are equally unfamiliar with System A. We believe that a quick learning curve is also one of the advantages of System A.
>
> In fact, before commencing the formal testing, we allowed them to play with the two systems. Furthermore, compared with related works (like Parsel), the two prompts mentioned in the problem are quite straightforward: System A's prompt merely asks the LLM to fill in the hole based on natural language, while System B's prompt provides users with a task format reference. Thus, the prompt plays a minimal role in the performance gap between the two systems. See the results on the relationship between user experience and performance below in Question 1.
>
> > ARC is designed as a test of human intelligence... Ideally the results would have included an additional dataset designed for synthesis (such as MBPP or HumanEval)
>
> We should clarify that what we are measuring is the system's ability to convey the user's intent through interaction, not the system's ability to solve problems on its own (e.g. the user has a correct algorithmic solution in mind, but it is tedious to realize it in Python). In this regard, ARC's features, which are easy for humans to understand and solve but difficult to express via programming, would be more suitable.
> Datasets such as HumanEval and MBPP are much simpler in comparison to ARC, with many questions solvable using just a few lines of Python code (the average line of code of MBPP, HumanEval, APPS, and ARC are 6.71, 7.78, 19.95, and 26.47 respectively).
>
> To show ANPL's generalization ability, we conducted an additional experiment on the APPS dataset and tested the capabilities of ANPL in implementing long programs. The results show that ANPL outperforms the original GPT-3.5-turbo and Parsel on APPS and can be applied to real-world applications with complicated control/data flows and long programs including text editor, robot controller, naive MAGIC card game, and LS-8 CPU. Specifically, for 20 APPS-interview tasks in human study, ANPL achieves 100%, GPT-3.5-turbo achieves 85%, and Parsel achieves 55%. More details are shown in the Global Rebuttal (1) (2) (3). Also, we explained the choice of ARC in the Global Rebuttal (4).
>
> ## Questions
>
> > ..results for users aggregated by the language model code generation experience?
>
> It is no longer feasible to find another group of users with rich experience in LLM code generation, but from the perspective of usage experience and problem-solving rate, we get the following table.
>
> | LLM code generation experience | System A | System B | System C | System D  |
> |--------------------------------|----------|----------|----------|-----------|
> |1|68.52%|63.08%|24.87%|17.90%|
> |2|71.57%|53.77%|21.49%|19.22%|
> |3|84.39%|51.82%|40.35%|21.49%|
> |4|63.64%|50.38%|13.04%|0.00%|
> |5|80.65%|45.16%|11.43%|14.29%|
>
> It is shown that there is no significant relationship between the user's LLM code generation experience and solving rate. Sometimes, users with an experience level of 1 can achieve higher accuracy compared to those with an experience level of 5. Therefore, user experience does not significantly influence the final accuracy. Additionally, we conducted a comprehensive 440-hour human study, which, to the best of our knowledge, is exceedingly thorough and the most intricate human study experiment within the field related to LLMs. Taking all these factors into consideration, we believe that this experiment is reliable.
>
> > The performance of System B appears to saturate very quickly. ... Were the contexts for the LLMs reset at any point for either ANPL or System B?
>
> The performance of System B saturates very quickly because it is difficult for LLMs to understand the user intent through chat. LLMs cannot accurately identify where the understanding gap between itself and the user arises. And this is precisely the significance of ANPL - ANPL allows users to interact effectively with LLM in a manner similar to debugging, ensuring that the user intent is almost always understood by LLM as the interaction progresses. Therefore, ANPL's performance improves continuously as user interactions go on.
>
> In fact, in the experiment, users were free to delete or modify their chat content and they can even write an ANPL program and then "compile" it by hand when using System B. In doing so, we ensured that the comparative experiment between System A and B was fair. We will revise the paper to make this clearer.
>
> We hope the rebuttal is adequate in addressing some of your concerns, and if so, please update your review in accordance.

---

> > ### Comment · Reviewer_tgZy · 2023-08-12
> >
> > Thanks for the response. I am satisfied with the choice of ARC as the dataset as well as the additional details on the experimental evaluation.
> >
> > I maintain my concerns about the core contributions: a programming system based on the combination of sketch-based synthesis with hand-engineered LLM prompts, neither of which is novel.
> >
> > I have therefore increased my score to a 5, but will not complain if the paper is ultimately accepted.

---

> > > ### Author Response · Authors · 2023-08-13
> > > **Thanks for your reply and here is the response to address your remaining concern**
> > >
> > > Thank you for increasing your score and further elaborating on the remaining concern!
> > >
> > > ### About the system innovation
> > > We conclude that our contribution as proposing **an interactive programming system** to significantly reduce programming complexity and assure code quality, where the users can iteratively interact with the system to **safely ground their minds to code**, is novel. And we believe that this kind of system is important to the development of the community by enabling users with limited programming experience to reliably program for user-specific demands with natural language in various domains including product design, robot control, etc. Additionally, the data generated in such a programming system (like the DARC dataset proposed in this paper, the ANPL code of APPS, ANPL projects like LS-8 CPU, etc.) can facilitate the advancement of relevant fields like program synthesis, human-ai interaction, and cognitive science.
> > >
> > > ### About the technique innovation
> > > **It is inappropriate to simply attribute ANPL to sketch + prompt**, as the scope of "sketch" is too broad and can encompass many things. A more proper understanding is that ANPL is based on (1) **the natural language programming paradigm** of separating control/data flow and sub-functions and (2) the corresponding **compiler**, which is designed to allow users to interact with the system and safely ground their intentions into the program. Techniques like AST parsing, code merging, and LLM prompting (only a simple prompt) are part of the compiler (Section 4).
> > >
> > > Specifically, for (1), we clarify that the programming paradigm of ANPL is **not similar to the ones before** and we merely leverage "sketch + holes" as a way of explaining our work in a language familiar to the community. "Sketch + holes" is a general philosophy adopted in program synthesis and **the key is to define the specific sketch and hole elements**. For example, the "sketch" in [1] represents the high-level natural language plan, the "sketch" in [2] represents the relevant part of the retrieved code, the "hole" in [3] represents unfinished code, and in SKETCH[4], the "hole" is an integer. **In our work**, the key insight is that control/data flows are important in conveying user intent with a small part of user effort while sub-functions stand in contrast. Thus, it is a sweet point for the trade-off between less user effort and more accurate user intent. **Under this insight**, we named the explicit control/data flows "sketch" and natural language sub-functions "holes" to make the idea easy to understand by the community.
> > >
> > > For (2), there might be a misunderstanding because prompt engineering is not the main part of the compiler. The compiler of ANPL consists of multiple important components including AST parsing, code merging, LLM prompting, and so on (Section 4), all of which work together to enable users to interact with ANPL and ground their thoughts to code.
> > >
> > > [1] Self-planning Code Generation with Large Language Models
> > >
> > > [2] SKCODER: A Sketch-based Approach for Automatic Code Generation
> > >
> > > [3] Representing partial programs with blended abstract semantics
> > >
> > > [4] Program synthesis by sketching
> > >
> > > We hope the response can address your concern and we will revise the paper to make this clear.

---

### Official Review · Reviewer_neh7 · 2023-07-07

**Soundness:** 2 fair
**Presentation:** 2 fair
**Contribution:** 2 fair
**Rating:** 6
**Confidence:** 4

**Summary:**

The paper proposes ANLP, a framework that interactively gets the definition of the data flow of a program from user input, and then fill the "holes" in the sketched program using LLMs. The authors conducted comprehensive human experiments on the Abstraction and Reasoning Corpus by letting human subjects solve the problems using an off-the-shelf language model and the ANPL-wrapper language model. The results show that with ANLP people are better at solving ARC problems.

**Strengths:**

1. Splitting the burden of reasoning between human programmers and language models is reasonable. Humans can focus on higher-level planning and leave the implementation details to language models.
2. The evaluation with human subjects is convincing. In terms of solving rate and efficiency, it's clear that ANPL is easier to use than bare ChatGPT.
3. Letting users define the dataflow in a python-like language is friendly for programmers and easy for the model to understand.

**Weaknesses:**

1. The idea of decomposing a task into modular functions is not new. As the authors have mentioned, even under the LLM setting it's been used by Parsel (https://arxiv.org/pdf/2212.10561.pdf). Even though the authors emphasizes the novelty of human interaction, I don't find your system significantly different from theirs. Of course your human evaluation is much more comprehensive.
2. Evaluating on ARC alone seems not enough. While the tasks seem good for testing the reasoning ability, it is not clear whether ANPL can work for real-world applications that help actual programmers. The definition of data flow can get complicated and harder for the model to understand as the program gets longer.

**Questions:**

1. Does ANPL work well for complicated data flows and longer programs?
2. Does ANPL work for real world applications at larger scales?

**Limitations:**

The authors have addressed the limitations.

---

> ### Author Rebuttal · Authors · 2023-08-09
>
> Thank you for your review comments! Your concerns were about the differences between ANPL and other similar work like Parsel, as well as the applicability of ANPL to a broader range of complicated real-world tasks. For the former, we have provided a detailed explanation and examples in our response. As for the latter, we have conducted an additional experiment on APPS, where ANPL achieves 100%, GPT-3.5-turbo achieves 85%, and Parsel achieves 55%. Please refer to the detailed responses below and the Global Rebuttal for more information.
>
> ## Weaknesses
>
> > 1.The difference from Parsel.
>
> Unlike Parsel which generates an implementation in Python from the problem description in 1-shot. ANPL allows the user to gradually clarify their intents through interactions. This enables the final code to converge gradually towards the correct direction.
> Specifically, compared with Parsel, we have the following distinctions:
>
> 1. **Control/data flows**. ANPL enables users to define explicit control/data flow and control flow while Parsel cannot due to the ambiguity in natural language (which indicates that our compiler design is entirely different). For example, we cannot extract the explicit data flow from the following Parsel code (randomly selected from the Parsel paper) because `tokens` is not defined as variables and we do not know the relationship between the three `tokens`:
>
>     ``` Python
>     tokenize(s): Convert a string into a list of tokens, including parens.
>     ... (omitted I/Os)
>     read_from_tokens(tokens): Translate tokens to their corresponding atoms, using parentheses for nesting lists.
>     ... (omitted I/Os)
>     ```
>     The natural language-defined control flow relies on LLM's code generation capabilities and thus its complexity is severely restricted. This makes Parsel challenging to extend to real-world applications with complicated control/data flows.
>     In comparison, ANPL allows users to define control/data flows either implicitly in natural language or explicitly in code, which is more flexible and can be applied to more complicated tasks.
>
> 2. **The debugging process**. Parsel faces difficulty in pinpointing bugs and modifying specific subfunctions due to the need for regenerating a whole new executable program for each change. ANPL, on the other hand, excels in precise adjustments without altering user-validated code. Debugging with LLMs, as discussed in Section 4, is complex due to unpredictable behavior, and resolving this hinges on ANPL's compiler implementation.
>
> A small experimental comparison between ANPL and Parsel can be found in the Global Rebuttal (1) (2), where for 20 APPS-interview tasks human study, ANPL achieves 100%, GPT-3.5-turbo achieves 85%, and Parsel can only achieve 55% because it is not designed for human interaction. In fact, whether in terms of design concepts or programming interfaces, ANPL and Parsel are not similar to each other. We suspect that the "resemblance" may arise because about 80% of both ANPL and Parsel codes are composed of natural language and function signatures which are the common parts among different natural language programming frameworks. A better approach to discerning the differences between the two is to compare their compiler algorithms. We will revise our paper to make this clearer.
>
> > 2.Evaluating on ARC alone seems not enough. ... The data flow can get complicated and harder for the model to understand as the program gets longer.
>
> Experimentally, we have conducted additional experiments on the APPS dataset and several long programming tasks to demonstrate ANPL's potential in real-world applications (with the longest programming consisting of 306 lines of Python). Details are shown in the Global Rebuttal (1) (2) (3). Also, we explained the choice of ARC in the Global Rebuttal (4).
> Briefly speaking, ANPL consistently outperforms baselines on the APPS experiment and can generalize to real-world applications like text editor, robot controller, naive MAGIC card game, and LS-8 CPU. We choose ARC because (a) ARC is much more difficult than MBPP and HumanEval and (b) what we are measuring is the system's ability to convey the user's intent through interaction, not the system's ability to solve problems on its own (e.g. the user has a correct algorithmic solution in mind, but it is tedious to realize it in Python).
>
> Theoretically, the complicated data flow does not make it difficult for the model to understand because the ANPL compiler parses data flow using parsing algorithms leveraging ASTs directly rather than relying on LLMs to generate the right code, which means that the data flow never needs to be "understood" by LLMs. This makes the (de)composition process much more stable, controllable, and extendable.
>
> ## Questions
>
> > 1.Does ANPL work well for complicated data flows and longer programs?
>
> Yes, it does. As stated in weakness 2, the complexity of the data control flow does not affect the compilation effectiveness of ANPL. In fact, the programs in ARC are already long compared with other datasets: The (min, max, mean) of lines of ARC's (passed) program is (4, 113, 26.47) while MBPP is (2, 50, 6.71), HumanEval is (2, 32, 7.78), and APPS is (1, 297, 19.95).
>
> Additionally, we have presented ANPL examples of programming tasks with complex control/data flows (Global Rebuttal (3) and Figure 2 in the pdf), all of which could be compiled and executed. The longest compiled Python code among them has 306 lines of Python code.
>
> > 2.Does ANPL work for real world applications?
>
> As mentioned above, we have conducted an additional human study on APPS dataset and we have implemented several representative long programming tasks to show ANPL's potential in real-world applications. However, we also acknowledge that ANPL, as a newly emerged programming framework, is fragile, so its development requires collective efforts from the community.
>
> We hope the rebuttal addresses some of your concerns. If so, please update your review in accordance.

---

> > ### Comment · Reviewer_neh7 · 2023-08-14
> >
> > Thank you for the response!
> >
> > I'm raising my score to 6.

---

> > > ### Author Response · Authors · 2023-08-15
> > >
> > > Thanks for raising your score! We are glad that the rebuttal has addressed some of your concerns!

---

### Official Review · Reviewer_NKWp · 2023-07-09

**Soundness:** 2 fair
**Presentation:** 3 good
**Contribution:** 2 fair
**Rating:** 6
**Confidence:** 3

**Summary:**

ANPL proposes a Sketch guided program synthesis using Natural Language in an interactive fashion.The results show better interactive coding than base LLM.

**Strengths:**

The paper shows potential in using sketch guided methods in conjunction with LM inference. It can be a good tool to use for debugging and potentially to teach programming.

**Weaknesses:**

Though the method show promise, some amount of work is necessary to justify the applicability of the system in wider settings. Some proof of guarantee on how the sketch helps in overall synthesis can also throw some light to the reader on why a system like this can be useful and should be adopted by wider research community. Some specific points:

1. Although an interesting direction, it would have been great to see the performance of the model in general purpose programming tasks, and not only for debugging.

2. The availability of only a specific syntax seem very limiting to applicability. It is not clear on how to use the system for new domains.

3. Some recent works on LLM that work in conjunction with Sketch guided synthesis seem to be missing.

**Questions:**

1. How do we extend the system to a general purpose program decomposition problem solving method?

**Limitations:**

The limitations section need considerable improvement.

---

> ### Author Rebuttal · Authors · 2023-08-09
>
> We thank you for your review comments! Your main concerns were about the generality of the programming framework and the adequacy of the references to related works.
> We have tried our best to address your concerns by incorporating additional experiments and adding related works. Please refer to the response below for more details.
> However, due to the brevity of your statement, some aspects may not have been fully understood, and we kindly request you provide a more detailed and specific elaboration of your points. This will help us improve the quality of the paper.
>
> ## Weaknesses
>
> > 1.Although an interesting direction, it would have been great to see the performance of the model in general purpose programming tasks, and not only for debugging.
>
> It should be noticed that ANPL **is a general-purpose programming framework** and can be applied to various tasks since it is a superset of Python (i.e. we can achieve anything that can be done with Python using ANPL). This is also stated in the first paragraph of Section 3.
>
> To provide more evidence of ANPL's generalizability, we conducted additional experiments related to the APPS dataset and several long programming tasks. Please see the details in the response (1) (2) (3) and the supplementary pdf of our Global Rebuttal.
> Briefly speaking, we conducted a 20 APPS tasks human study, in which ANPL achieves 100%, GPT-3.5-turbo achieves 85%, and Parsel achieves 55%. This shows the advantage of ANPL in general typical programming tasks.
> Also, we implemented several realistic and long projects including text editor, robot controller, naive MAGIC card game, and LS-8 CPU, which shows the generalizability of ANPL.
>
> We should clarify that ANPL is designed to grounding user intent to code through interaction which is different from code generation but also an important direction for LLM programming assistants because it is hard for LLMs and other code generation techniques to understand the user intent and generate correct code in 1-shot .
>
> It would be most helpful for us if you can elaborate on what you mean by "not only for debugging". If you can give us a few example tasks you have in mind, we'll be able to address your concerns better.
>
> > 2.The availability of only a specific syntax seem very limiting to applicability. It is not clear on how to use the system for new domains.
>
> ANPL is a general-purpose programming framework. Perhaps there was a misunderstanding caused by our experiments on the ARC dataset, as the input-output format for that task is similar (specific syntax). However, ANPL can be used to address any programming-related problems with Python. To demonstrate this, we have included additional experiments on the APPS dataset and conducted extensive experiments on long program development using ANPL (please refer to the experiments mentioned in the previous question and Global Rebuttal (1-3)).
>
> In addition, if we have any misunderstandings about the issue, please respond to us promptly, and we will provide the corresponding responses.
>
> > 3.Some recent works on LLM that work in conjunction with Sketch guided synthesis seem to be missing.
>
> Thanks for your reminder. We planned to add the following sketch-guided references to the related work section:
>
> [1] Self-planning Code Generation with Large Language Models
>
> [2] SKCODER: A Sketch-based Approach for Automatic Code Generation
>
> [3] Representing partial programs with blended abstract semantics
>
> Among them, [1] proposes to generate code with the planning phase and implementation phase, where LLM outputs solution steps in the planning phase and then generates the code step by step in the implementation phase. [2] retrieves similar code snippets from a codebase and then extracts the relevant code sketch to guide the following synthesis process. [3] represents unfinished code with the help of "hole" and approximates its semantics using neural networks to guide the synthesis by execution.
>
> Different from them, our work focuses on incorporating user interaction within the process of code generation by introducing a natural language programming framework, which allows users to program and debug to clarify their intent progressively. Although they have also employed the concepts of "sketch" and "hole", the specific definitions are different. In our work, the "sketch" represents the control/data flow written by users while the "hole" represents natural language. In theirs, for example, the "sketch" in [1] represents the high-level natural language plan, the "sketch" in [2] represents the relevant part of the retrieved code, and the "hole" in [3] represents unfinished code.
>
> Also, it would be helpful if you can point us to a few specific ones you have in mind for us to make a comparison.
>
> ## Questions
>
> > 1.How do we extend the system to a general purpose program decomposition problem solving method?
>
> As mentioned above, ANPL is already a general-purpose programming framework and can be applied to various tasks since it is a superset of Python. We have conducted an additional human study on APPS dataset and we have implemented several representative long programming tasks to show ANPL's potential in real-world applications (see Global Rebuttal (1-3)). One way to extend ANPL is to let LLM generate high-level plans with ANPL automatically, which can be seen as a kind of new prompting technique by prompting with control/data flows instead of purely natural language. We take it as our future work.
>
> We hope the rebuttal is adequate in addressing some of your concerns, and if so, please update your review in accordance.

---

> ### Author Response · Authors · 2023-08-18
> **Dear reviewer, we look forward to your response**
>
> Note that the discussion will be ended in 4 days, we kindly request reviewer NKWp engage in the discussion. During the rebuttal period, we carried out additional experiments and are here to ensure that we have addressed all questions or doubts regarding to general programming tasks, new domains, and related work. We hope everything is clear and the recent experiments have addressed the concerns. Thanks!

---

> > ### Comment · Reviewer_NKWp · 2023-08-22
> > **Thank you for your response**
> >
> > Thank you for responding to the doubts. I have carefully read the author response and re-read the manuscript. The particularities of the presentation of the paper made it a bit unclear on what exactly the contribution of the paper was. It was unclear that the authors have proposed a general purpose solution, which was my main concern.
> >
> > Overall the contribution of this paper is good and after careful consideration I have modified my initial score. I still urge the authors to add some additional insights in the conclusion of the paper and expanding on the limitations with focus on future extensions.

---

> > > ### Author Response · Authors · 2023-08-22
> > >
> > > We thank the reviewer for the response! We will revise our paper carefully to make our contributions, limitations, and future extensions clearer (the detailed plan can be seen in our response to reviewer ecZq)!

---

### Author Rebuttal · Authors · 2023-08-09

We thank the reviewers and ACs for their time and effort.

We are glad the reviewers acknowledge the following contributions:novel and reasonable for proposing a LLM programming system by introducing extensive (i.e. non 1-shot)user interactions(neh7, tgZy, ecZq); comprehensive human study and strong empirical results (neh7, tgZy, ecZq), and clear presentations (tgZy).

In this response, we will summarize and address the important concerns raised:

(1) The generalization ability of ANPL to different domains.

(2) Comparison of ANPL against Parsel (https://arxiv.org/pdf/2212.10561.pdf), a non-interactive LLM-backed compiler that also compiles natural language to code.

(3) Can ANPL compile "realistic" and "long" programs effectively?

(4) Why did we choose ARC as our benchmark?

(5) A minor misunderstanding on the "sketches and holes" program synthesis paradigm.
### (1) The generalization ability of ANPL and (2) Comparison against Parsel

We conduct a small study with 4 participants and 20 randomly selected APPS-interview problems. Other domains considered were MBPP, HumanEval, and APPS-competition problems. We did not choose MBPP and HumanEval because they are much simpler compared with ARC, with many questions solvable using just a few lines of Python code (The [min, max, mean] of lines of ARC's (passed) program is [4, 113, 26.47] while MBPP is [2, 50, 6.71], HumanEval is [2, 32, 7.78]). Also, we did not test on competition-level tasks limited by user ability. Each participant was instructed to solve each problem with three systems (ANPL, GPT-3.5-turbo, and Parsel) appearing in a random order for a total of 30 minutes per system. We replicated Parsel with GPT-3.5-turbo because Codex is no longer available.

As shown in supplemented pdf Figure 1:

(a) ANPL outperforms the original GPT-3.5-turbo and Parsel in final solving rate and solving time which is consistent with the result on ARC. Specifically, ANPL achieves 100% solving rate, GPT-3.5-turbo achieves 85%, and Parsel achieves 55%. Also, the 1-shot performance (without interaction) of them is 30%, 0%, and 15% respectively.

(b) The solving rate for APPS-interview is higher than the one on ARC, indicating APPS-interview is easier than ARC for human interaction. In fact, the challenge of APPS lies in how users solve the tasks rather than grounding users' intents to code.

(c) Parsel performs the worst because Parsel and ANPL have different design objectives: ANPL is designed to grounding user intent to code through interaction while Parsel is designed for automatic reasoning without human intervention. Specifically, Parsel does not even retain the context of past interactions. This leaves the only method for "debugging" is to continuously modify the initial Parsel code and provide more I/Os until it can be compiled to Python in a single attempt.

Overall, we conclude that
(a) ANPL is generalizable to different domains and
(b)
ANPL and Parsel are substantially different systems: ANPL is designed as an interactive programming system to significantly reduce programming complexity and assure code quality, where the users can iteratively interact with the system to safely ground their minds to code; While Parsel is primarily a reasoning system, where the programming task is given to it "as is" and it leverages the capabilities of LLM to decompose and ground it to code without any user interventions.
### (3) Can ANPL compile "realistic" and "long" programs effectively?

To further show the ANPL's generalization ability, we conduct a qualitative study by using ANPL to compile several projects drawn from CS courses and GitHub: text editor, robot control, naive MAGIC card game, and LS-8 CPU. These tasks are the prototypes of realistic applications and are representative in terms of the program form with complex control flows and many sub-functions. They are much longer (198 lines of code on average) than existing program synthesis benchmarks (MBPP[6.71], HumanEval[7.78], APPS[19.95])

It is shown in pdf Figure 2 that ANPL can be used to implement these applications with much less coding effort (with 33.21% lines of code) and obviously lower programming barrier (by using natural language) than Python. Also, the current system including GPT-4 and Parsel cannot solve these problems easily in our test. Such results show that ANPL has the potential to be adopted in real scenarios with great superiority of lower programming complexity and better code quality assurance.
### (4) Why did we choose ARC as our benchmark?

We clarify that what we are measuring is the system's ability to convey the user's intent through interaction, not the system's ability to solve problems on its own (e.g. the user has a correct algorithmic solution in mind, but it is tedious to realize it in Python). In this regard, ARC's features (easy for humans to understand and solve but difficult to express via programming) would be more suitable than typical programming datasets such as MBPP and HumanEval.

We find that typical programming datasets are simpler for programming compared with ARC. The average line of code of MBPP, HumanEval, APPS, and ARC are 6.71, 7.78, 19.95, and 26.47 respectively.
Competition datasets like APPS challenge the users to solve the tasks rather than how to generate code that fulfills users' intents.

### (5) A minor misunderstanding of the "sketches and holes" program synthesis paradigm.

Sketches + holes is a classical program synthesis paradigm and we are in no way claiming that as a contribution. We merely leverage it as a way of explaining our work in a language familiar to the community.  We will make this point clearer in our draft.
We will make this point clearer in our draft.

We hope our responses have convincingly addressed all reviewer's concerns by conducting supplementary experiments and giving detailed explanations. Please do not hesitate to let us know of any additional comments.

---

### Decision · Program_Chairs · 2023-09-21

**Decision:**

Accept (poster)

**Comment:**

This submission presents ANLP, a programming system that mediates between a user and a backend LLM through a sketching approach. Rather than focusing on generating programs in a single step, this system aims to help users express high-level ideas as executable code. A user interface helps users to sketch a high-level data flow with holes, where each hole is annotated with a natural language description. Initial solutions can be tuned by inspecting the inputs and output of each hole individually, allowing for example to specify IO constraints for each hole.

Reviewers agree that the proposed method is useful to help users program, especially for debugging or teaching purposes, and that it improves over directly using an LLM.

However, reviewers also raised the issue of a somewhat confusing presentation: many misunderstandings arose from the writing and required clarification during the rebuttal phase, though these mainly require smaller changes to the text. A bigger issue is that even though the programming system is the core contribution, its user interface is only discussed in the supplement at this time. I would encourage the authors to think about how to make it easier for readers to understand what the user interface actually looks like (e.g., what is the first thing a user sees when trying to solve a new problem).

The fact that the utility of the system was evaluated using a substantial user study was appreciated by reviewers, but questions remained on the chosen dataset (the Abstraction and Reasoning Corpus). During the rebuttal phase, the authors provided additional experiments on the more standard APPS dataset, involving longer programs. I will note that personally, I believe that the problem decomposition dataset obtained during the user study (DARC) is a particularly important contribution.

Overall, I agree with the reviewers that this is an interesting paper that deserves presentation at NeurIPS.